# Performative Power

**Moritz Hardt**[*]
Max-Planck Institute for Intelligent Systems, Tübingen
hardt@is.mpg.de

**Meena Jagadeesan**[*]
UC Berkeley
mjagadeesan@berkeley.edu

**Celestine Mendler-Dünner**[*]
Max-Planck Institute for Intelligent Systems, Tübingen
cmendler@tuebingen.mpg.de

## Abstract

We introduce the notion of performative power, which measures the ability of a firm operating an algorithmic system, such as a digital content recommendation platform, to cause change in a population of participants. We relate performative power to the economic study of competition in digital economies. Traditional economic concepts struggle with identifying anti-competitive patterns in digital platforms not least due to the complexity of market definition. In contrast, performative power is a causal notion that is identifiable with minimal knowledge of the market, its internals, participants, products, or prices.

We study the role of performative power in prediction and show that low performative power implies that a firm can do no better than to optimize their objective on current data. In contrast, firms of high performative power stand to benefit from steering the population towards more profitable behavior. We confirm in a simple theoretical model that monopolies maximize performative power. A firm's ability to personalize increases performative power, while competition and outside options decrease performative power. On the empirical side, we propose an observational causal design to identify performative power from discontinuities in how digital platforms display content. This allows to repurpose causal effects from various studies about digital platforms as lower bounds on performative power. Finally, we speculate about the role that performative power might play in competition policy and antitrust enforcement in digital marketplaces.

## 1 Introduction

Digital platforms pose a well-recognized challenge for antitrust enforcement. Traditional market definitions, along with associated notions of competition and market power, map poorly onto digital platforms. A core challenge is the difficulty of precisely modeling the interactions between the market participants, products, and prices. An authoritative report, published by the Stigler Committee [2019], details the many challenges associated with digital platforms, among them: *"Pinpointing the locus of competition can also be challenging because the markets are multisided and often ones with which economists and lawyers have little experience. This complexity can make market definition another hurdle to effective enforcement."* Published the same year, a comprehensive report from the European Commission calls for *"less emphasis on analysis of market definition, and more emphasis on theories of harm and identification of anti-competitive strategies."* [Crémer et al., 2019]

Our work responds to this call by developing a normative and technical proposal for reasoning about power in digital economies, while relaxing the reliance on market definition. Our running

---

[*]Authors listed in alphabetical order.

36th Conference on Neural Information Processing Systems (NeurIPS 2022).

example is a digital content recommendation platform. The platform connects content creators with viewers, while monetizing views through digital advertisement. Key to the business strategy of a firm operating a digital content recommendation platform is its ability to predict revenue for content that it recommends. Often framed as a supervised learning task, the firm trains a statistical model on observed data to predict some proxy of revenue, such as clicks, views, or engagement. Better predictions enable the firm to more accurately identify content of interest and thus increase profit.

A second way of increasing profit is more subtle. The platform can use its predictions to steer participants towards modes of consumption and production that are easier to predict and monetize. For example, the platform could reward consistency in the videos created by content creators, so that the popularity of their videos becomes more predictable. Similarly, the platform could recommend addictive content to viewers, appealing to behavioral weaknesses to drive up viewer engagement. How potent such a strategy is depends on the extent to which the firm is able to steer participants, which we argue reveals a salient power relationship between the platform and its participants.

## 1.1 Our contribution

We introduce the notion of *performative power* that quantifies a firm's ability to steer a population of participants. We argue that the sensitivity of participant behavior to algorithmic changes in the platform provides an important indicator of the firm's power. Performative power is a causal statistical notion that directly quantifies how much participants change in response to actions by the platform, such as updating a predictive model. In doing so it avoids market specifics, such as the number of firms involved, products, and monetary prices. Neither does it require a competitive equilibrium notion as a reference point. Instead, it focuses on where rubber meets the road: the algorithmic actions of the platform and their causal powers.

We build on recent developments in performative prediction [Perdomo et al., 2020] to articulate the difference between learning and steering in prediction. We show that under low performative power, a firm cannot do better than standard supervised learning on observed data. Intuitively, this means the firm optimizes its loss function *ex-ante* on data it observes. We interpret this optimization strategy as analogous to the firm being a price-taker, an economic condition that arises under perfect competition in classical market models. We contrast this optimization strategy with a firm that performs *ex-post* optimization and takes advantage of performative power to achieve lower expected risk.

To better understand our definition, we study performative power in the concrete algorithmic market model of strategic classification. Strategic classification models participants as best-responding agents that change their features rationally in response to a predictor to achieve a better prediction outcome. We study the role of different economic factors by extending the standard model to incorporate competing firms and outside options. Our first observation is that a *monopoly* firm can derive significant performative power because participants are willing to incur a cost up to the utility of using the service in order to adjust to the firm's predictor. Moreover, performative power is maximized if the firm can personalize decisions to individual users. Our second observation is that performative power decreases in the presence of *competition* and *outside options*. When firms compete for participants, offering services that are perfect substitutes for each other, then even two firms can lead to zero performative power. This result stands in analogy with classical Bertrand competition.

On the empirical side, we propose a causal design to identify performative power in the context of a recommender system arranging content into display slots. This design, we call *discrete display design (DDD)*, establishes a connection between performative power and the causal effect of display position on consumption. To derive a lower bound on performative power, DDD constructs a hypothetical algorithmic action that aggregates the causal effects of display position across the population. This allows us to repurpose reported causal effects of display position as lower bounds on performative power. It also charts out a concrete empirical strategy for measuring power in digital economies.

Finally, we examine the role of performative power in competition policy. We contrast performative power with traditional measures of market power, describe how performative power can capture complex behavioral patterns, and discuss the potential role of performative power in antitrust debates.

## 1.2 Related work

Our notion of performative power builds on the development of performativity in prediction by Perdomo et al. [2020]. Performativity captures that the predictor can influence the data-generating

process, a dependency ruled out by the traditional theory of supervised learning. A growing line of work on performative prediction, e.g., [Mendler-Dünner et al., 2020; Drusvyatskiy and Xiao, 2022; Izzo et al., 2021; Dong and Ratliff, 2021; Miller et al., 2021; Brown et al., 2022; Li and Wai, 2021; Ray et al., 2022; Jagadeesan et al., 2022; Wood et al., 2022], has studied different optimization challenges and solution concepts in performative prediction. Rather than viewing performative effects as a challenge for the learning algorithm, we argue that performativity reveals a salient power relationship between the decision maker and the population. From an optimization perspective, our work demonstrates that sufficiently high performative power is necessary for performative optimization approaches to be beneficial compared with standard supervised learning.

The *strategic classification* setup we use for our case study was proposed in [Brückner et al., 2012; Hardt et al., 2016] and is closely related to a line of work in the economics community [Frankel and Kartik, 2022; Ball, 2020; Hennessy and Goodhart, 2020; Frankel and Kartik, 2019]. A line of work on strategic classification makes the assumption that performative effects are the result of individuals manipulating their features so as to best respond to the deployment of a predictive model. The focus has been on describing participant behavior in response to a single firm acting in isolation. Our extensions incorporate additional market factors into the model, such as outside options or the choice between competing firms, which we believe are helpful for gaining a better understanding of interactions in digital economies. Beyond the case of a single classifier, recently, Narang et al. [2022] and Piliouras and Yu [2022] consider multiple firms simultaneously applying retraining algorithms in performative environments and analyze convergence to equilibrium solutions. Ginart et al. [2021] study another model of feedback loops arising from competition between machine learning models.

There is extensive literature on the topic of competition on digital platforms that we do not attempt to survey here. For starting points, see, for example, recent work by Bergemann and Bonatti [2022], a survey by Calvano and Polo [2021], a discussion by Parker et al. [2020], the reports already mentioned [Stigler Committee, 2019; Crémer et al., 2019], as well as a macroeconomic perspective on the topic [Syverson, 2019].

## 2 Performative power

Fix a set $\mathcal{U}$ of participants interacting with a designated firm, where each $u \in \mathcal{U}$ is associated with a data point $z(u)$. Fix a metric $\mathrm{dist}(z, z')$ over the space of data points. Let $\mathcal{F}$ denote the set of actions a firm can take. We think of an action $f \in \mathcal{F}$ as a predictor that the firm can deploy at a fixed point in time. For each participant $u \in \mathcal{U}$ and action $f \in \mathcal{F}$, we denote by $z_f(u)$ the potential outcome random variable representing the counterfactual data of participant $u$ if the firm were to take action $f$.

**Definition 1** (Performative Power). *Given a population $\mathcal{U}$, an action set $\mathcal{F}$, potential outcome pairs $(z(u), z_f(u))$ for each unit $u \in \mathcal{U}$ and action $f \in \mathcal{F}$, and a metric* $\mathrm{dist}$ *over the space of data points, we define the* performative power *of the firm as*

$$\mathrm{P} := \sup_{f \in \mathcal{F}} \frac{1}{|\mathcal{U}|} \sum_{u \in \mathcal{U}} \mathbb{E}\left[\mathrm{dist}\left(z(u), z_f(u)\right)\right],$$

*where the expectation is over the randomness in the potential outcomes.*

The expression inside the supremum generalizes an average treatment effect, corresponding to scalar valued potential outcomes and the absolute value as metric. We could generalize other causal quantities such as heterogeneous treatment effects, but this avenue is not subject of our paper. The definition takes a supremum over possible actions a firm can take at a specific point in time. We can therefore lower bound performative power by estimating the causal effect of any given action $f \in \mathcal{F}$.

Having specified the sets $\mathcal{F}$ and $\mathcal{U}$, estimating performative power amounts to causal inference involving the potential outcome variables $z_f(u)$ for unit $u \in \mathcal{U}$ and action $f \in \mathcal{F}$. In an observational design, an investigator is able to identify performative power without an experimental intervention on the platform. We propose and apply one such observational design in Section 5. In an experimental design, the investigator deploys a suitably chosen action to estimate the effect. Neither route requires understanding the specifics of the market in which the firm operates. It is not even necessary to know the firm's objective function, how it optimizes its objective, and whether it successfully achieves its objective. In practice, the dynamic process that generates the potential outcome $z_f(u)$ may be highly complex, but this complexity does not enter the definition. Consequently, the definition applies to complex multisided digital economies that defy mathematical specification. To make this abstract concept of performative power more concrete, let us instantiate it in a concrete example.

**Example: Digital content recommendation.** Consider a digital content recommendation platform, such as the video sharing services YouTube or Twitch. The platform aims to recommend channels that generate high revenue, personalized to each viewer. Towards this goal, the platform collects data to build a predictor $f$ for the value of a channel $c$ to a viewer with preferences $p$. Let $x = (x_c, x_p)$ be the features used for the prediction task that capture attributes $x_c$ of the channel and the attributes $x_p$ of the viewer preferences. Let $y$ be the target variable, such as *watch time*, that acts as a proxy for the monetary value of showing a channel to a specific viewer. For concreteness, take the supervised learning loss $\ell(f(x), y)$ incurred by a predictor $f$ to be the squared loss $(f(x) - y)^2$.

When defining performative power, participants could either be viewers or content creators. The definition is flexible and applies to both. By selecting the units $\mathcal{U}$, which features to include in the data point $z$, and how to specify the distance metric $\mathrm{dist}$, we can pinpoint specific power relationships.

*Content creators.* The predictor $f$ can affect the type of videos that content creators stream on their channels. For example, content creators might strategically adjust various features of their content relevant for the predicted outcome, such as the length, type or description of their videos, to improve their ranking. Thus, by changing how it predicts the monetary value of a channel, the platform can induce changes in the content on the channel. To measure this source of power, we let the participants $\mathcal{U}$ be content creators and suppose that each content creator $u \in \mathcal{U}$ maintains a channel of videos. Let the data point $z(u)$ correspond to features $x_c$ characterizing the channel $c$ created by content creator $u$. Let $\mathrm{dist}$ be a metric over features of content. The resulting instantiation of performative power measures the changes in content induced by potential implementations $\mathcal{F}$ of the prediction function. In Section 4, we investigate this form of performative power from a theoretical perspective by building on the setup of *strategic classification*.

*Viewers.* The predictor $f$ can shape the consumption patterns of viewers, since viewers tend to follow recommendations when deciding what content to consume (e.g. [Ursu, 2018]). Thus, by changing which content it recommends to a user, the platform can induce changes in the target variable: how much time the users spends watching content on a given channel. Let's suppose that we wish to investigate the effect of the predictor on viewer consumption of a certain genre of content (e.g. radical content). To formalize this source of power, we let the units $\mathcal{U}$ be viewers. Let the data point $z(u)$ correspond to how long the viewer $u$ spends watching content in the genre of interest. More formally, let $z(u)$ for a user $u$ with preferences $p$ be equal to the cumulative watch time over pairs $(x_c, x_p)$ where $c$ is a channel within the genre of interest. Let $\mathrm{dist}(z, z') = |z - z'|$ capture the difference in watch time. The resulting instantiation of performative power measures the changes in consumption of a given genre of content induced by a set of prediction functions $\mathcal{F}$. In Section 5, we propose an *observational design* to identify this quantity.

## 3 Learning versus steering

Performative power enters the firm's optimization problem and has direct consequences for their optimization strategies. Instead of identifying the best action $f$ while treating data as fixed, high performative power enables the firm to *steer* the population towards data that it prefers. In the following, we elucidate the role of performative power in optimization and study the equilibria attained in an economy of performative predictors.

### 3.1 Optimization strategies

Let us focus on predictive accuracy as the optimization objective of the firm. Hence, the goal of the firm is to choose a predictive model $f$ that suffers small loss $\ell(f(x), y)$ measured over instances $(x, y)$. To make the role of steering explicit we distinguish between the *ex-ante* loss $\ell(f(x(u)), y(u))$ and the *ex-post* loss $\ell(f(x_f(u)), y_f(u))$. The former describes the loss that the firm can optimize when building the predictor. The latter describes the loss that the firm observes after deploying $f$. More formally, the *ex-post risk* that the firm suffers after deploying $f$ on a population $\mathcal{U}$ is given by

$$\frac{1}{|\mathcal{U}|} \sum_{u \in \mathcal{U}} \ell(f(x_f(u)), y_f(u)). \tag{1}$$

Expression (1) is an instance of what Perdomo et al. [2020] call *performative risk* of a predictor. That is the loss a predictor incurs on the distribution over instances it induces. To simplify notation we adopt their conceptual device of a distribution map: $\mathcal{D}(\theta)$ maps a predictive model, characterized by model parameters $\theta$, to a distribution over data instances.

To express the firm's optimization problem within the framework of performative prediction, we define a data instance as $z(u) = (x(u), y(u))$ for $u \in \mathcal{U}$ so we can capture performativity in the features as well as in the labels. Let the firm's action correspond to choosing a parameter vector $\theta$ for its predictor $f_\theta$ from an action set $\mathcal{F} = \Theta$. Then, the *aggregate distribution* over data $\mathcal{D}(\theta)$ corresponds to the distribution over the potential outcome variable $z_\theta(u)$ after the firm takes action $\theta \in \Theta$, where the randomness comes from $u$ being uniformly drawn from $\mathcal{U}$ as well as randomness in the potential outcomes. The firm's ex-post risk (1) for deploying $f_\theta$ is the performative risk:

$$\mathrm{PR}(\theta) := \mathop{\mathbb{E}}_{z \sim \mathcal{D}(\theta)}[\ell(\theta; z)],$$

where the loss typically corresponds to the mismatch between the predicted label and the true label: $\ell(\theta; z) = \ell(f_\theta(x), y)$ for $z = (x, y)$.

Observe that $\theta$ arises in two places in the objective: in the distribution $\mathcal{D}(\theta)$ and in the loss $\ell(\theta; z)$. Thus, for any choice of model $\phi$, we can decompose the performative risk $\mathrm{PR}(\theta)$ as:

$$\mathrm{PR}(\theta) = \mathrm{R}(\phi, \theta) + (\mathrm{R}(\theta, \theta) - \mathrm{R}(\phi, \theta)) \tag{2}$$

where $\mathrm{R}(\phi, \theta) := \mathbb{E}_{z \sim \mathcal{D}(\phi)} \ell(\theta; z)$ denotes the loss of a model $\theta$ on the distribution $\mathcal{D}(\phi)$. This tautology highlights the difference between learning and steering and we differentiate between the following two optimization approaches:

**Ex-ante optimization.** Ex-ante optimization focuses on optimizing the first term in the decomposition (2). For any $\phi$, the resulting minimizer can be computed statistically and corresponds to $\theta_{\mathrm{SL}} := \arg\min_{\theta \in \Theta} \mathrm{R}(\phi, \theta)$. Let $f_\phi$ be any previously chosen model, then employing supervised learning on historical data sampled from $\mathcal{D}(\phi)$ corresponds to what we call ex-ante optimization.

**Ex-post optimization.** In contrast to ex-ante optimization, *ex-post optimization* accounts for the impact of the model on the distribution. It trades-off the two terms in (2), and directly optimizes the performative risk $\theta_{\mathrm{PO}} := \arg\min_{\theta \in \Theta} \mathrm{PR}(\theta)$. Solving this problem exactly, and finding the performative optimum $\theta_{\mathrm{PO}}$ requires optimization over the distribution map $\mathcal{D}(\theta)$.

In the context of digital content recommendation, ex-ante optimization corresponds to training $\theta$ on historical data collected by the platform, whereas ex-post optimization selects $\theta$ based on randomized experiments, A/B testing or explicit modeling of $\mathcal{D}(\theta)$. It holds that $\mathrm{PR}(\theta_{\mathrm{PO}}) \leq \mathrm{PR}(\theta_{\mathrm{SL}})$, because in ex-post optimization the firm can choose to steer the population towards more predictable behavior.

**Remark** (Generalizing to other objectives)**.** *We thus far have focused on the firm's prediction problem when describing performative effects. Nonetheless, the conceptual distinction between learning and steering applies to general optimization objectives. Ex-ante optimization corresponds to optimizing on historical data, whereas ex-post optimization corresponds to optimizing over the counterfactuals.*

### 3.2 Gain of ex-post optimization is bounded by a firm's performative power

We show that the gain of ex-post optimization over ex-ante optimization can be bounded by the firm's performative power with respect to the set of actions $\Theta$ and the data vector $z = (x, y)$. Intuitively, if the firm's performative power is low, then the distributions $\mathcal{D}(\theta)$ and $\mathcal{D}(\phi)$ for any $\theta, \phi \in \Theta$ are close to one another. Coupled with a regularity assumption on the loss, this means that the second term in (2) should be small. Thus, using the ex-ante approach of minimizing the first term produces a near-optimal ex-post solution, as we demonstrate in the following result:

**Proposition 1.** *Let* $\mathrm{P}$ *be the performative power of a firm with respect to the action set* $\Theta$*. Let* $L_z$ *be the Lipschitzness of the loss in* $z$ *with respect to the metric* dist*. Let* $\theta_{\mathrm{PO}}$ *be the ex-post solution and* $\theta_{\mathrm{SL}}$ *be the ex-ante solution computed from* $\mathcal{D}(\phi)$ *for any past deployment* $\phi \in \Theta$*. Then, we have that*

$$\mathrm{PR}(\theta_{\mathrm{SL}}) \leq \mathrm{PR}(\theta_{\mathrm{PO}}) + 4L_z\mathrm{P}.$$

The gain achievable through ex-post optimization is bounded by performative power. Hence, a firm with small performative power cannot do much better than ex-ante optimization and might be better off sticking to classical supervised learning practices instead of engaging with ex-post optimization.

### 3.3 Ex-post optimization in an economy of predictors

The result in Proposition 1 studies the optimization strategy of a single firm in isolation. In this section, we investigate the interaction between the strategies of multiple firms that optimize simultaneously over the same population. We consider an idealized marketplace where $C$ firms

all engage in ex-post optimization and we assume all exogenous factors remain constant. Let $\mathcal{D}^C(\theta^1, \ldots, \theta^{i-1}, \theta^i, \theta^{i+1}, \ldots, \theta^C)$ be the distribution over $z(u)$ induced by each firm $i \in [C]$ deploying model $f_{\theta^i}$. Let $\ell_i$ denote the loss function chosen by firm $i$. We say a set of predictors $[f_{\theta^1}, \ldots, f_{\theta^C}]$ is a *Nash equilibrium* if and only if no firm has an incentive to unilaterally deviate from their predictor using ex-post optimization:

$$\theta^i \in \underset{\theta \in \Theta}{\arg\min} \; \underset{z \sim \mathcal{D}^C(\theta^1, \ldots, \theta^{i-1}, \theta, \theta^{i+1}, \ldots, \theta^C)}{\mathbb{E}} [\ell_i(\theta; z)].$$

First, we show that at the Nash equilibrium, the suboptimality of each predictor $f_{\theta^i}$ on the induced distribution depends on the performative power of the respective firm.

**Proposition 2.** *Suppose that the economy is in a Nash equilibrium $(\theta^1, \ldots, \theta^C)$, and firm $i$ has performative power $\mathrm{P}_i$ with respect to the action set $\Theta$. Let $L_z$ be the Lipschitzness of the loss $\ell_i$ in $z$ with respect to the metric* dist. *Then, it holds that $\mathbb{E}_{z \sim \mathcal{D}}[\ell_i(\theta^i; z)] \leq \min_\theta \mathbb{E}_{z \sim \mathcal{D}}[\ell_i(\theta; z)] + L_z \mathrm{P}_i$, where $\mathcal{D} = \mathcal{D}^C(\theta^1, \ldots, \theta^C)$ is the distribution induced at the equilibrium.*

Proposition 2 implies that if the performative power of all firms is small ($\mathrm{P}_i \to 0 \; \forall i$), then the equilibrium becomes indistinguishable from that of a static, non-performative economy with distribution $\mathcal{D}$ over content. Interestingly, there is an important distinction with the static setting: if the firms were to collude—for example, because of common ownership [Azar et al., 2018] —then they would be able to significantly shift the distribution. In particular, even if the performative power $\mathrm{P}_i$ of any given firm $i$ is small, the aggregate performative power of a set of firms $S$ can be much larger.

To illustrate this, we consider a *mixture economy*, where all of the firms share a common loss function $\ell$ and performative power is uniformly distributed across firms (we formalize the construction of a mixture economy in Appendix B.1). We can apply Proposition 2 to analyze the equilibria as $C \to \infty$.

**Corollary 1.** *Suppose that all firms $i \in [C]$ share the same loss function $\ell_i = \ell$. Let $\theta^*$ be a symmetric Nash equilibrium in the mixture economy with $C$ platforms. As $C \to \infty$, it holds that:*

$$\underset{z \sim \mathcal{D}^C(\theta^*, \ldots, \theta^*)}{\mathbb{E}} [\ell(\theta^*; z)] \; \to \; \min_\theta \; \underset{z \sim \mathcal{D}^{C=1}(\theta^*)}{\mathbb{E}} [\ell(\theta; z)].$$

Corollary 1 demonstrates that a symmetric equilibrium approaches a *performatively stable point* of the monopoly economy where $C = 1$ as the number of firms in the economy grows and the performative power of each individual firm diminishes. In contrast, if all $C$ firms collude, they would obtain the performative power of a monopoly platform and instead choose a *performatively optimal point* of $\mathcal{D}^{C=1}$. A competitive economy can thus exhibit a very different equilibrium from that of the monopoly or collusive economy.

## 4 Performative power in strategic classification

We now turn to a stylized market model and investigate how performative power depends on the economy in which the firm operates. Specifically, we use *strategic classification* [Hardt et al., 2016] as a test case for our definition. In strategic classification, participants strategically adapt their features with the goal of achieving a favorable classification outcome. Hence, performative power is determined by the degree to which a firm's classifier can impact participant features. We use this concrete market setting to examine the qualitative behavior of performative power in the presence of competition and outside options.

### 4.1 Strategic classification setup

Let $x(u)$ be the *features* and $y(u)$ the *binary label* describing a participant $u \in \mathcal{U}$. A firm chooses a binary predictor $f : \mathbb{R}^m \to \{0, 1\}$ and incurs loss $\ell(f(x), y) = |f(x) - y|$. Let $\mathcal{D}_{\text{orig}}$ denote the base distribution over features and labels $(x_{\text{orig}}(u), y_{\text{orig}}(u))$ absent any strategic adaptation, which we assume is continuous and supported everywhere. Let $\mathcal{D}(f)$ be the distribution over potential outcomes $(x_f(u), y_f(u))$ that arises from the response of participant $u$ to the deployment of a model $f$. We assume that participant $u$ incurs a cost $c(x_{\text{orig}}(u), x')$ for changing their features to $x'$. In line with the standard setup, the cost for feature changes is measured relative to the *original* features. We assume $c$ is a metric and any feature change that deviates from the original features results in nonnegative cost for participants. Further, we assume the label does not change, i.e., $y_f(u) = y_{\text{orig}}(u)$.

Since performative effects surface as changes in participant features, we measure performative power over $z(u) = x(u)$. In Appendix B.2, we describe how this instantiation of performative power can be interpreted in the context of content recommendation platforms and hiring platforms.

## 4.2 Performative power in the monopoly setting

Consider an economy with a single firm that offers utility $\gamma > 0$ to its participants for a positive prediction. Participants want to use the service regardless of what classifier the firm chooses. In addition, assume there is an *outside options* at utility level $\beta > 0$ that does not demand any extra effort from participants. This decreases the budget they are willing to invest to their *surplus utility* $\Delta\gamma = \max(0, \gamma - \beta)$. We adopt the following standard rationality assumption on participant behavior.

**Assumption 1** (Participant Behavior Specification). *Let $\Delta\gamma \geq 0$ be the surplus utility that a participant can expect from a positive classification outcome from classifier $f$ over any outside option. Then, a participant $u \in \mathcal{U}$ with original features $x_{\mathrm{orig}}(u)$ will change their features according to $x_f(u) = \arg\max_{x'} (\Delta\gamma f(x') - c(x_{\mathrm{orig}}(u), x'))$.*

As a result of Assumption 1, a participant will change their features if and only if the cost of a feature change is no larger than $\Delta\gamma$. Furthermore, if participants change their features, then they will expend the minimal cost required to achieve a positive outcome. This specification of participant behavior allows us to bound the firm's performative power in terms of the cost function $c$ and the distance function $\mathrm{dist}$. Namely, the set of potential values $x_f(u)$ can take on is restricted to $\mathcal{X}_{\Delta\gamma}(u) := \{x : c(x_{\mathrm{orig}}(u), x) \leq \Delta\gamma\}$. Thus, the effect of a change in the decision rule on participant $u$ can be upper bounded by the distance between $x(u)$ and the most distant point in $\mathcal{X}_{\Delta\gamma}$. Aggregating these unilateral effects yields a bound on performative power:

**Lemma 1.** *The performative power $\mathrm{P}$ of the firm with respect to any set of predictors $\mathcal{F}$ and a population $\mathcal{U}$ can be upper bounded as*

$$\mathrm{P} \leq \frac{1}{|\mathcal{U}|} \sum_{u \in \mathcal{U}} \sup_{x' \in \mathcal{X}_{\Delta\gamma}(u)} \mathrm{dist}(x(u), x'). \tag{3}$$

**Tightness of Lemma 1.** If the firm action space $\mathcal{F}$ is restricted to a parametric family of predictors, the upper bound in Lemma 1 need not be tight. A typical decision rule, such as a linear threshold classifier, does not impact all participants $u \in \mathcal{U}$ equally. More specifically, the amount of change that the firm can induce with a decision rule $f$ on an individual $u$ depends the relative position of their features $x_{\mathrm{orig}}(u)$ to the decision boundary. (In Appendix B.3 we provide a precise bound for threshold classifiers).

Interestingly, the ability to *personalize* decisions to each user maximizes performative power.

**Proposition 3.** *Consider a population $\mathcal{U}$ of users. Let $\mathcal{F}$ be the set of all functions from $\mathbb{R}^m$ to $\{0, 1\}$. Suppose that the first coordinate of $x(u)$ is immutable, and uniquely identifies user $u \in \mathcal{U}$: that is, $x_{\mathrm{orig}}(u)[1] = x_f(u)[1] = i$ for all $f \in \mathcal{F}$. Then, the performative power of the firm matches the upper bound in (3).*

The intuition behind Proposition 3 is that personalization enables the firm to simultaneously extract the maximum utility from each individual participant.

**Role of surplus utility $\Delta\gamma$.** Let us now investigate the role of $\Delta\gamma$ in the upper bound (3). Recall that user behavior is determined by the cost $c$ of changing features relative to $x_{\mathrm{orig}}(u)$, performative power is measured as the distance from the current state $x(u)$ with respect to $\mathrm{dist}$. The Lipschitz constant $L := \sup_{x,x'} \frac{\mathrm{dist}(x,x')}{c(x,x')}$ allows us to relate the two metrics and derive a simpler upper bound:

**Corollary 2.** *The performative power of a firm in the monopoly setup can be bounded as $\mathrm{P} \leq 2L\,\Delta\gamma$, where $\Delta\gamma$ measures the surplus utility offered by the service of the firm over outside options.*

Corollary 2 makes explicit that $\Delta\gamma > 0$ is a prerequisite for a firm to have any performative power. This qualitative behavior is in line with common intuition in economics that monopoly power relies on the firm offering a service that is superior to existing options (i.e. $\gamma > \beta$).

### 4.3 Firms competing for participants

We next consider a model of *competition* between two firms where participants always choose the firm that offers higher utility. We demonstrate how the presence of competition reduces the performative power of a firm. In particular, we will show that for a natural constraint on the firm's action set, each firm's performative power can drop to zero at equilibrium, regardless of how much utility participants derive from the firm's service.

To model competition in strategic classification, we specify participant behavior as follows: Given that the first firm deploys $f_1$ and the second firm deploys $f_2$, then participant $u$ will choose $f_1$ if $\max_{x'} (f_1(x') - c(x_{\mathrm{orig}}(u), x')) > \max_{x'} (f_2(x') - c(x_{\mathrm{orig}}(u), x'))$, and choose $f_2$ otherwise. We assume that a participant tie-breaks in favor of the lower threshold, randomizing if the two thresholds are equal. After choosing firm $i \in \{1, 2\}$, they change their features according to Assumption 1 as $x_f(u) = \arg\max_{x'} (\gamma f_i(x') - c(x_{\mathrm{orig}}(u), x'))$, where $\gamma$ is the utility of a positive outcome.

We assume that the firm chooses their classifier based on the following utility function. For a rejected participant, the firm receives utility $0$ and for an accepted participant, the firm receives utility $\alpha > 0$ if they have a positive label and utility $-\alpha$ if they have a negative label. We assume that the firm's action set is constrained to models for which it derives non-negative utility. More specifically, if $f_\theta$ denotes the model deployed by the competing firm, let the action set $\mathcal{F}^+(\theta)$ of this firm denote the set of models that yield non-negative utility for the firm.

For simplicity, focus on a 1-dimensional setup where $\mathcal{F}$ is the set of threshold functions. We assume that the cost function $c(x, x')$ is continuous in both of its arguments, strictly increasing in $x'$ for $x' > x$, strictly decreasing for $x' < x$, and satisfies $\lim_{x' \to \infty} c(x, x') = \infty$. Furthermore, we assume that the posterior $p(x) = \mathbb{P}_{\mathcal{D}_{\mathrm{orig}}}[Y = 1 \mid X = x]$ is strictly increasing in $x$ with $\lim_{x \to -\infty} p(x) = 0$, and $\lim_{x \to \infty} p(x) = 1$.

We show that the presence of competition can drive the performative power of each firm to zero.

**Proposition 4.** *Consider the 1-dimensional setup with two competing firms specified above. Suppose that the economy is at a symmetric Nash equilibrium $(\theta^*, \theta^*)$. If $L < \infty$, then the performative power of either firm with respect to the action set $\mathcal{F}^+(\theta^*)$ is $\mathrm{P} = 0$.*

The intuition behind Proposition 4 is that performative power of a firm purely arises from how much larger the current threshold $\theta$ is than the minimum threshold a firm can deploy within their action set $\mathcal{F}^+(\theta)$. At the Nash equilibrium (where both firms best-respond with respect to their utility functions taking their own performative effects into account), the firms deploy exactly the minimum threshold within their action set. The formal proof of the result can be found in Appendix C.9.

Corollary 4 bears an intriguing resemblance to well-known results on market power under Bertrand competition in economics (see e.g., [Baye and Kovenock, 2008]) that show how a state of zero power is reached with only two competing firms.

## 5 Discrete display design

Now that we have examined the theoretical properties of performative power, we turn to the question of *measuring* performative power from observational data. We focus on our running example of digital content recommendation and we propose an observational design to measure the recommender system's ability to shape consumption patterns.

### 5.1 The causal effect of position

We assume that there are $C$ pieces of content $\mathcal{C} = \{0, 1, 2, \ldots, C - 1\}$ that the platform can present in $m$ display slots. We make the convention that item $0$ corresponds to leaving the display slot empty. We focus on the case of two display slots ($m = 2$) since it already encapsulates the main idea. The first display slot is more desirable as it is more likely to catch the viewer's attention. For example, the first display slot could be the first ad slot on a page of search results. Researchers have studied the causal effect of position on consumption, often via quasi-experimental methods such as regression discontinuity designs, but also through experimentation in the form of A/B tests.

**Definition 2** (Causal effect of position)**.** *Let the treatment $T \in \{0, 1\}$ be the action of flipping the content in the first and second display slots for a viewer $u$, and let the potential outcome variable $Y_t(u)$*

*indicate whether, under the treatment $T = t$, viewer $u$ consumes the content that is initially in the first display slot. We call the corresponding average treatment effect*

$$\beta = \left| \mathbb{E} \left[ Y_1 - Y_0 \right] \right|$$

*the causal effect of position, where the expectation is taken over the population of viewers and the randomness in the potential outcomes.*

For example, Narayanan and Kalyanam [2015] estimate the causal effect of position in search advertising, where advertisements are displayed across a number of ordered slots whenever a keyword is searched. They measured the causal effect of position on click-through rate of participants.

## 5.2   From causal effect of position to performative power

The identification strategy we propose, called *discrete display design (DDD)*, derives a lowerbound on performative power by repurposing existing measures of the causal effect of position. Note that we focus on content recommendation in this section, the design however can be generalized to other settings where the firm's action corresponds to a discrete decision of how to display content. Setting up the DDD involves two steps: First, we need to instantiate the definition of performative power with a suitable action set so that the firm's actions result in swapping the position of content items, and second, we plug in the causal effect of position to lower bound performative power.

While the first step is mostly a technical exercise, the second step relies on a crucial assumption. In particular, it involves relating the unilateral causal effect of position to performative power that quantifies the effect of an action on the entire population of viewers. Thus, for being able to extrapolate the effect from a single viewer to the population DDD relies on a non-interference assumption. In the advertising example, this means that the ads shown to one viewer do not influence the consumption behavior of another viewer. Let us investigate the two steps in greater detail:

**Step 1: Instantiating performative power.**   Let the units $\mathcal{U}$ be the set of viewers. For each viewer $u \in \mathcal{U}$ let $z(u) \in \mathbb{R}^C$ be the distribution over content items $\mathcal{C}$ consumed by viewer $u$, represented as a histogram. More formally, let $z(u)$ be a vector in the $C$-dimensional probability simplex where the $i$th coordinate is the probability that viewer $u$ consumes content item $i$. The metric $\text{dist}(z, z')$ is the $\ell_1$-distance $\text{dist}(z, z') = \sum_{i=0}^{C-1} |z[i] - z'[i]|$.

The decision space $\mathcal{F}$ of the firm corresponds to its decisions of how to arrange content in the $m = 2$ display slots. It is natural to decompose this decision into a continuous score function $s$ followed by a discrete conversion function $\kappa$ that maps scores into an allocation. The score function $s \colon \mathcal{U} \to \mathbb{R}^C$ maps the viewer to a vector of scores, where each coordinate is an estimate of the quality of the match between the viewer and the corresponding piece of content. The conversion function $\kappa \colon \mathbb{R}^C \to \mathcal{C}^2$ takes as input the vector of scores and outputs an ordered list of items with the top 2 scores. We assume the platform displays these 2 items in order and the conversion function $\kappa$ is fixed. Hence, we identify the firm's action space with the set of feasible score functions $\mathcal{S} \subseteq \mathcal{U} \to \mathbb{R}^C$.

To define the reference state $z(u)$, we think of $s_{\text{curr}}$ as being the score function currently deployed by the platform. Let $\delta$ be the maximum difference in the highest score and second highest score for any user under $s_{\text{curr}}$. Consider the set $\mathcal{S}$ of scoring functions defined as

$$\mathcal{S} := \left\{ s \colon \mathcal{U} \to \mathbb{R}^C \mid \forall u \in \mathcal{U} \colon \|s(u) - s_{\text{curr}}(u)\|_\infty \leq \delta \right\}.$$

Notably, there exists an $s_{\text{swap}} \in \mathcal{S}$ that is capable of swapping the order of the first and second highest scoring item under $s_{\text{curr}}$ for any user $u \in \mathcal{U}$ simultaneously. We denote the counterfactual variable corresponding to a score function $s \in \mathcal{S}$ as $z_s(u)$. Given this specification, performative power with respect to the action set $\mathcal{S}$ can be bounded by the causal effect of $s_{\text{swap}}$ as follow

$$\mathrm{P} = \sup_{s \in \mathcal{S}} \frac{1}{\mathcal{U}} \sum_{u \in \mathcal{U}} \|z_{s_{\text{curr}}}(u) - z_s(u)\|_1 \geq \frac{1}{\mathcal{U}} \sum_{u \in \mathcal{U}} \|z_{s_{\text{curr}}}(u) - z_{s_{\text{swap}}}(u)\|_1. \tag{4}$$

**Step 2: Lower bounding performative power.**   To relate the lower bound on performative power from (4) to the causal effect of position, let $i_{\text{top}}(u) = \kappa \circ s_{\text{curr}}(u)[1]$ denote the coordinate of the item displayed to user $u$ in the first display slot under $s_{\text{curr}}$. Then, we can lower bound each term in the sum (4) as $\|z_{s_{\text{curr}}}(u) - z_{s_{\text{swap}}}(u)\|_1 \geq |z_{s_{\text{curr}}}(u)[i_{\text{top}}(u)] - z_{s_{\text{swap}}}(u)[i_{\text{top}}(u)]|$. Now to enable

us to study the effect of changing $s_{\text{curr}}$ to $s_{\text{swap}}$ independently for each user we place the following non-interference assumption on the counterfactual variables which closely relates to the stable unit treatment value assumption (SUTVA) [Imbens and Rubin, 2015] prevalent in causal inference.

**Assumption 2** (No interference across units). *For any $u \in \mathcal{U}$ and any pair of scoring functions $s_1, s_2 \in \mathcal{S}$, if $\kappa(s_1(u)) = \kappa(s_2(u))$, it also holds that $z_{s_1}(u) = z_{s_2}(u)$.*

The assumption requires that there are no spill-over or peer effects and the content a viewer consumes only depends on the content recommended to them and not the content recommended to other viewers. The effect of a unilateral change to the consumption of item $i_{\text{top}}(u)$ under $s_{\text{swap}}$ exactly corresponds to what we defined as the causal effect of position. Aggregating these unilateral causal effects across all viewers in the population we obtain a lowerbound on performative power. The proof of Theorem 1 can be found in Appendix C.12.

**Theorem 1.** *Let $\mathrm{P}$ be performative power as instantiated above. If Assumption 2 holds, then performative power is at least as large as the causal effect of position*

$$\mathrm{P} \geq \beta.$$

Let us return to the search advertisement marketplace study of Narayanan and Kalyanam [2015] to demonstrate how we can leverage Theorem 1 to relate the findings of their observational causal design to performative power. In particular, they examine position effects in search advertising, where ads are displayed across a number of ordered slots whenever a keyword is searched. They found that the effect of showing an ad in display slot 1 versus display slot 2 corresponds to 0.0048 clicks per impression (see Table 2 in their paper). By treating each incoming keyword query as a distinct "viewer" $u$, this number exactly corresponds to what we defined as the causal effect of position. Thus, we can apply Theorem 1 to get $P \geq 0.0048$. Putting this into context; the mean click-through rate in display slot 2 is 0.023260. Hence, the lower bound 0.0048 is a 21% percent increase relative to the baseline. The firm thus has a substantial ability to shape what advertisements users click on.

## 6 Discussion

We discuss the potential role of performative power in competition policy and antitrust enforcement.

The complexity of digital marketplaces has made it necessary to develop new approaches for evaluating and regulating these economies. One challenge is that traditional measures of market power—such as the Lerner Index [Lerner, 1934], or the Herfindahl–Hirschman Index (HHI)—are based on classical markets for homogeneous goods, but these markets map poorly to digital economies. In particular, these measures struggle to appropriately capture the multi-sided nature of digital economies and to account for the role of behavioral weaknesses of consumers—such as tendencies for single-homing, vulnerability to addiction, and the impact of framing and nudging on participant behavior [e.g. Thaler and Sunstein, 2008; Fogg, 2002]. We further expand on this discussion in Appendix A.

By focusing on observable statistics, performative power could be particularly helpful in markets that resist a clean specification. Performative power is sensitive to the market nuances without explicitly modeling them. For example, suppose that as a result of uncertainty about market boundaries, a regulator failed to account for a competitor in a marketplace. Performative power would still implicitly capture the impact of the competitor and indicate the market is more competitive than suspected.

We leave open the question of how to best instantiate performative power in a given marketplace. Conceptually, we view performative power as a tool to flag market situations that merit further investigation, since it corresponds to "potential for harm to users". However, if a regulator wishes to draw fine-grained conclusions about consumer harm, it is crucial to appropriately instantiate the choice of action set $\mathcal{F}$, the definition of a population $\mathcal{U}$, and the specification of the features $z$. As an example, we show in Appendix B.4 how to closely relate performative power into consumer harm for strategic classification. In general, however, harm and power are two distinct normative concepts, and going from performative power to consumer harm thus requires additional substantive arguments.

## Acknowledgments

We're grateful to Guy Aridor, Dirk Bergemann, Emilio Calvano, Gabriele Carovano, and Martin C. Schmalz for helpful feedback and pointers.

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
