# A  Performative power in competition policy

We expand on the discussion from Section 6 about the potential role of performative power in competition policy and antitrust enforcement as a tool complementary to existing machinery.

## A.1  Measures of market power in economics

Typical measures of market power in economic theory focus on classical pricing markets of homogeneous goods, where a firm's primary action is choosing a price to sell the good or the quantity of the good to sell. The scalar nature of these quantities enables them to be easily compared across different market contexts and firms. In addition, the utility of the firm and the utility of participants are inversely related: a higher price yields greater utility for the firm and lower utility for all participants. This simple relationship allows one to directly reason about participant welfare and profit of firms. However, the situation of a digital economy is more complex [Stigler Committee, 2019; Crémer et al., 2019]. As a result, classical measures from competition theory that primarily focused on price effects struggle to accurately characterize these economies. Let us illustrate these challenges using two textbook definitions of market power.

**Lerner index.** The Lerner index [Lerner, 1934] quantifies the pricing power of a firm, measuring by how much the firm can raise the price above marginal costs. Marginal costs reflect the price that would arise in a perfectly competitive market. A major issue of applying this standard definition of market power in the context of digital economies is that it is not clear what the competitive reference state should look like, "We have lost the competitive benchmark,"[2] as Jacques Crémer said. Thus, measures based on profit margin cannot directly be adopted as a proxy for market power in these settings.

**Market share.** Measures such as the Herfindahl–Hirschman index (HHI), which is used by the US federal trade commission[3] to measure market competitiveness, is based on *market share*: the fraction of participants who participate in a given firm. However, the validity of market share as a proxy for power relies on a specific model of competition where the elasticity of demand is low. This model is challenging to justify[4] in the context of digital economies where opening an account on a platform is very simple and usually free of charge. In addition, not all participants with accounts on a digital platform are equally active and inactive participants should not factor into the market power of a firm the same way active participants do. Market share is not sufficiently expressive to make this distinction.

In contrast, performative power is a causal notion of influence that does not require a precise specification of the market but is still sensitive to the nuances of the market. The definition therefore could be serve as a useful tool in markets that resist a clean mathematical specification.

## A.2  Complex consumer behavior

Participant behavior plays a critical role in digital market places. As outlined by the Stigler Committee [2019], "the findings from behavioral economics demonstrate an under-recognized market power held by incumbent digital platforms." In particular, behavioral aspects of consumers—such as tendencies for single-homing, vulnerability to addiction, and as well as the impact of framing and nudging on participant behavior [e.g. Thaler and Sunstein, 2008; Fogg, 2002]—can be exploited by firms in digital economies, but do not factor into traditional measures of market power. By focusing on changes in participant features, performative power has the potential to capture the effects of these behavioral patterns while again sidestepping the challenges of explicitly modeling them.

## A.3  From performative power to consumer harm

Performative power focuses on measuring power rather than harm. The relationship to harm depends on the choice and interpretation of the outcome variable and requires additional substantive arguments.

---

[2]Opening statement at the 2019 Antitrust and competition conference – digital platforms, markets, and democracy

[3]See https://www.justice.gov/atr/herfindahl-hirschman-index (retrieved January, 2022).

[4]This critique is similar to the disconnect between the Cournot model and the Bertrand model in classical economics [Bornier, 1992]. E.g., "concentration is worse than just a noisy barometer of market power" [Syverson, 2019].

High performative power and the ability of a firm to steer participants naturally bears the potential for user harm, whenever there exists a misalignment between the utilities of the platform and the participants. Recent work by Shmueli and Tafti [2020] backs a similar concern with empirical evidence. However, in general, harm and power are two fundamentally distinct normative concepts. Performative power does not necessarily imply harm, but it can serve as an indicator of *potential* harm that can help flag market situations that merit further investigations by regulators. In fact, the report of the European Commission on competition policy for the digital era [Crémer et al., 2019] suggests in several contexts where regulators should be suspicious of power and suggests putting burden of proof on the firm rather than the regulators.

In specific contexts, we can establish an exact correspondence between performative power and harm. By carefully choosing the parameters in the definition, performative power can be instantiated to more closely capture manifested harm to a particular (sub-)population of participants we wish to investigate. We implement this principle in Section 4 where we establish a connection between performative power and the utility of users. More generally, this connection can be achieved if the attributes $z(u)$ consists of the sensitive features that are impacted by the firm, the distance function is aligned with the utility function of participants, and the set $\mathcal{F}$ reflects actions that are taken by the firm.

# B  Additional results and discussions for Section 3 and Section 4

## B.1  Formalization of a mixture economy

We formalize the construction of a mixture economy. Let $z(u), z_\theta^{C=1}(u)$ denote the pair of counterfactual outcomes before and after the deployment of $\theta$ in a hypothetical monopoly economy where a single firm holds all the performative power. Let $\mathcal{D}^{C=1}(\theta)$ be the distribution map associated with the variables $z_\theta^{C=1}(u)$ for $u \in \mathcal{U}$. In a uniform mixture economy, we assume that each participant $u \in \mathcal{U}$ uniformly chooses between the $C$ firms. The counterfactual $z_\theta(u)$ associated with one firm changing its predictor to $\theta$ is equal to $z(u)$ with probability $1 - 1/C$ and $z_\theta^{C=1}(u)$ otherwise.

## B.2  Interpretation of performative power in setup in Section 4

In our running example of digital content recommendations where participants correspond to content creators, the instantiation of performative power in Section 4 measures how much the content of each channel changes with changes in the recommendation algorithm. As a different example, suppose that firm is a hiring platform such as HireVue that uses video data from interviews to a compute a performance prediction for an applicant for a job. In this case, the participants would correspond to applicants, and the data vector would correspond to the applicant's voice patterns and interview responses.

When instantiating the definition of performative power, the choice of distance metric enables us to define how to weight specific feature changes. In our content recommendation example, if we are interested in the burden on *content creators*, we choose the distance metric to be aligned with the cost function $c$ of producing a piece of content. However, if we are interested in measuring the impact of changes in content on viewers, a distance metric that reflects harm to viewers might be more appropriate. We keep this distance metric abstract in our analysis.

## B.3  Example: Monopoly power in heterogeneous setting

Different participants are typically impacted differently by a classifier, depending on their relative position to the decision boundary, as visualized in Figure 1(a). As a result of this heterogeneity, the upper bound in Lemma 1 is not necessarily tight, because the firm can not extract the full utility from all participants simultaneously.

Let us investigate the effect of heterogeneity in a concrete 1-dimensional setting where $\text{dist}_X(x, x') = c(x, x') = |x - x'|$. Consider a set of actions $\mathcal{F}$ that corresponds to the set of all threshold functions. Suppose that the posterior $p(x) = \mathbb{P}[Y = 1 \mid X = x]$ satisfies the following regularity assumptions: $p(x)$ is strictly increasing in $x$ with $\lim_{x \to -\infty} p(x) = 0$, and $\lim_{x \to \infty} p(x) = 1$. Now, let $\theta_{\text{SL}}$ be the supervised learning threshold, which is the unique value where $p(\theta_{\text{SL}}) = 0.5$. We can then obtain the following bound on the performative power P with respect to any $\mathcal{F}$ assuming the firm's classifier is

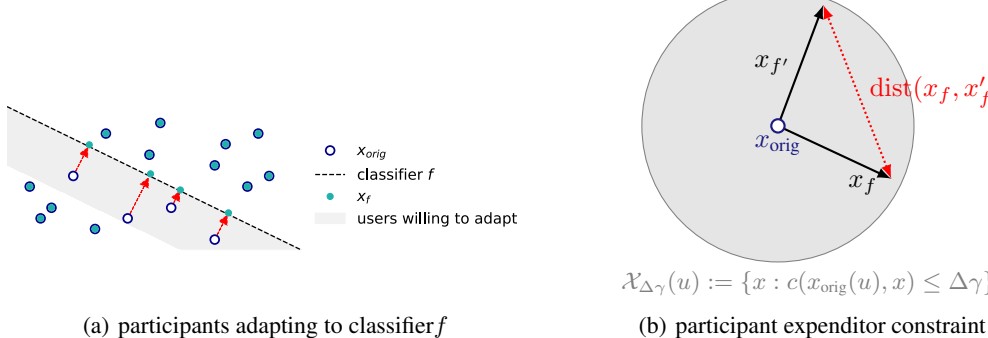

(a) participants adapting to classifier $f$

(b) participant expenditor constraint

Figure 1: Illustrations for 2-dimensional strategic classification example

$\theta_{\text{SL}}$ in the current economy (see Proposition 5):

$$0.5\Delta\gamma \mathop{\mathbb{P}}_{\mathcal{D}_{\text{orig}}} \left[ x \in [\theta_{\text{SL}}, \theta_{\text{SL}} + 0.5\Delta\gamma] \right] \leq \text{P} \leq \Delta\gamma. \tag{5}$$

This bound illustrates how performative power in strategic classification depends on the fraction of participants that fall in between the old and the new threshold. As long as the density in this region is non-zero, a platform that offers $\Delta\gamma > 0$ utility will also have strictly positive performative power, providing a lower bound on P.

**Proposition 5.** *Suppose that* $\text{dist}(x, x') = c(x, x') = |x - x'|$. *Let us consider a set of actions* $\mathcal{F}$ *that corresponds to the set of all threshold functions. Suppose that the posterior* $p(x) = \mathbb{P}[Y = 1 \mid X = x]$ *satisfies the following regularity assumptions:* $p(x)$ *is strictly increasing in $x$ with* $\lim_{x \to -\infty} p(x) = 0$, *and* $\lim_{x \to \infty} p(x) = 1$. *Now, let* $\theta_{\text{SL}}$ *be the supervised learning threshold, which is the unique value where* $p(\theta_{\text{SL}}) = 0.5$. *If the firm's classifier is $\theta_{SL}$ in the current economy, then performative power* P *with* $\mathcal{F}$ *can be bounded as:*

$$0.5\gamma \mathop{\mathbb{P}}_{\mathcal{D}_{orig}} \left[ x \in [\theta_{\text{SL}}, \theta_{\text{SL}} + 0.5\Delta\gamma] \right] \leq \text{P} \leq 2\Delta\gamma. \tag{6}$$

### B.4  Optimization strategies, performative power, and user harm

The fact that a monopoly firm has nonzero performative power has consequences for the optimization strategies that it would use, as we discussed in Section 3. To make this explicit, let's contrast the solutions of ex-ante and ex-post optimization in a simple one-dimensional setting.

**Example 1** (1-dimensional setting). *Consider a 1-dimensional feature vector $x \in \text{R}$ and suppose that the posterior* $p(x) = \mathbb{P}[Y = 1 \mid X = x]$ *is strictly increasing in $x$ with* $\lim_{x \to -\infty} p(x) = 0$, *and* $\lim_{x \to \infty} p(x) = 1$. *Now consider a set of actions* $\mathcal{F}$ *that corresponds to the set of all threshold functions and set* $\text{dist}(x, x') = c(x, x') = |x - x'|$. *Let* $\theta_{\text{SL}}$ *be the supervised learning threshold from ex-ante optimization, which is the unique value where* $p(\theta_{\text{SL}}) = 0.5$. *Then, the ex-post threshold lies at* $\theta_{\text{PO}} = \theta_{\text{SL}} + \Delta\gamma$.

In Example 1 ex-post optimization leads to a higher acceptance threshold than ex-ante optimization. Thus, for any setting where the participants utility is decreasing in the threshold (e.g., the class of utility functions that Milli et al. [2019] call *outcome monotonic*), this implies that ex-post optimization creates stronger negative externalities for participants than ex-ante optimization. Furthermore, the effect grows with the performative power of the firm. In the extreme case of the monopoly setting with no outside options, ex-post optimization can leave certain participants with a net utility of 0 and thus can transfer the entire utility from these participants to the firm.

## C  Proofs

### C.1  Auxiliary results

The proofs for Section 3 leverage the following lemma, which bounds the diameter of $\Theta$ with respect to Wasserstein distance in distribution map.

**Lemma 2.** *Let* $P$ *be the performative power with respect to* $\Theta$*. For any* $\theta, \theta' \in \Theta$*, it holds that* $\mathcal{W}(\mathcal{D}(\theta), \mathcal{D}(\theta')) \leq 2P$.

*Proof.* Let $\theta_{\mathrm{curr}}$ be the current classifier weights. We use the fact that for any weights $\theta'' \in \Theta$, it holds that $\mathcal{W}(\mathcal{D}(\theta_{\mathrm{curr}}), \mathcal{D}(\theta'')) \leq \frac{1}{|U|} \sum_{u \in \mathcal{U}} \mathbb{E}[\mathrm{dist}(z(u), z_{\theta''}(u))]$ where the expectation is over randomness in the potential outcomes. This follows from the definition of Wasserstein distance—in particular that we can instantiate the mass-moving function by mapping each participant to themselves. Thus, we see that:

$$\mathcal{W}(\mathcal{D}(\theta), \mathcal{D}(\theta')) \leq \mathcal{W}(\mathcal{D}(\theta), \mathcal{D}(\theta_{\mathrm{curr}})) + \mathcal{W}(\mathcal{D}(\theta_{\mathrm{curr}}), \mathcal{D}(\theta'))$$

$$\leq \frac{1}{|U|} \sum_{u \in \mathcal{U}} \mathbb{E}[\mathrm{dist}(z(u), z_\theta(u))] + \frac{1}{|U|} \sum_{u \in \mathcal{U}} \mathbb{E}[\mathrm{dist}(z(u), z_{\theta'}(u))]$$

$$\leq 2 \sup_{\theta'' \in \Theta} \frac{1}{|U|} \sum_{u \in \mathcal{U}} \mathbb{E}[\mathrm{dist}(z(u), z_{\theta''}(u))]$$

$$\leq 2P,$$

where the last line uses the definition of performative power that bounds the effect of any $\theta$ in the action set $\Theta$ on the participant data $z$. $\square$

### C.2 Proof of Proposition 1

Let $\phi$ be the previous deployment inducing the distribution on which the supervised learning threshold $\theta_{\mathrm{SL}}$ is computed. Let $\theta^*$ be an optimizer of $\min_{\theta \in \Theta} \mathrm{R}(\theta_{\mathrm{PO}}, \theta)$, where we recall the definition of the decoupled performative risk as $\mathrm{R}(\phi, \theta) := \mathbb{E}_{z \sim \mathcal{D}(\phi)} \ell(\theta; z)$. Then, we see that for any $\phi$:

$$\mathrm{PR}(\theta^{\mathrm{SL}}) - \mathrm{PR}(\theta_{\mathrm{PO}})$$
$$= \left( \mathrm{R}(\theta^{\mathrm{SL}}, \theta^{\mathrm{SL}}) - \mathrm{R}(\phi, \theta^{\mathrm{SL}}) \right) + \mathrm{R}(\phi, \theta^{\mathrm{SL}}) - \mathrm{R}(\theta_{\mathrm{PO}}, \theta_{\mathrm{PO}})$$
$$\leq \left( \mathrm{R}(\theta^{\mathrm{SL}}, \theta^{\mathrm{SL}}) - \mathrm{R}(\phi, \theta^{\mathrm{SL}}) \right) + \mathrm{R}(\phi, \theta^*) - \mathrm{R}(\theta_{\mathrm{PO}}, \theta^*)$$
$$\leq L_z \mathcal{W}(\mathcal{D}(\theta^{\mathrm{SL}}), \mathcal{D}(\phi)) + L_z \mathcal{W}(\mathcal{D}(\phi), \mathcal{D}(\theta_{\mathrm{PO}}))$$
$$\leq 4 L_z P.$$

The first inequality follows because $\theta^*$ minimizes risk on the distribution $\mathcal{D}(\theta_{\mathrm{PO}})$, while $\theta^{\mathrm{SL}}$ minimizes risk on $\mathcal{D}(\phi)$. The second inequality follows from the dual of the Wasserstein distance where $L_z$ is the Lipschitz constant of the loss function in the data argument $z$. The last inequality follows from Lemma 2.

Now, suppose that $\ell$ is $\gamma$-strongly convex. Then we have that:

$$\mathrm{R}(\theta, \theta_{\mathrm{PO}}) - \mathrm{R}(\theta, \theta_{\mathrm{SL}}) \geq \frac{\gamma}{2} \|\theta_{\mathrm{PO}} - \theta_{\mathrm{SL}}\|^2$$

Again applying Lemma 2,

$$\mathrm{PR}(\theta_{\mathrm{SL}}) = \mathrm{R}(\theta_{\mathrm{SL}}, \theta_{\mathrm{SL}})$$
$$\leq \mathrm{R}(\theta, \theta_{\mathrm{SL}}) + L_z \mathcal{W}(\mathcal{D}(\phi), \mathcal{D}(\theta_{\mathrm{SL}}))$$
$$\leq \mathrm{R}(\phi, \theta_{\mathrm{SL}}) + 2 L_z P$$
$$\leq \mathrm{R}(\phi, \theta_{\mathrm{PO}}) - \frac{\gamma}{2} \|\theta_{\mathrm{PO}} - \theta_{\mathrm{SL}}\|^2 + 2 L_z P$$
$$\leq \mathrm{R}(\theta_{\mathrm{PO}}, \theta_{\mathrm{PO}}) + L_z \cdot \mathcal{W}(\mathcal{D}(\phi), \mathcal{D}(\theta_{\mathrm{PO}})) - \frac{\gamma}{2} \|\theta_{\mathrm{PO}} - \theta_{\mathrm{SL}}\|^2 + 2 L_z P$$
$$\leq \mathrm{PR}(\theta_{\mathrm{PO}}) + 4 L_z P - \frac{\gamma}{2} \|\theta_{\mathrm{PO}} - \theta_{\mathrm{SL}}\|^2.$$

Using that $\mathrm{PR}(\theta_{\mathrm{PO}}) \leq \mathrm{PR}(\theta_{\mathrm{SL}})$, we find that

$$\frac{\gamma}{2} \|\theta_{\mathrm{PO}} - \theta_{\mathrm{SL}}\|^2 \leq 4 L_z P.$$

Rearranging gives

$$\|\theta_{\mathrm{PO}} - \theta_{\mathrm{SL}}\| \leq \sqrt{\frac{8 L_z P}{\gamma}}.$$

### C.3 Proof of Proposition 2

Let's focus on firm $i$, fixing classifiers selected by the other firms. Let's take $\mathrm{PR}$ and $\mathrm{R}$ to be defined with respect to $\mathcal{D}(\cdot) = \mathcal{D}(\theta_1, \ldots, \theta_{i-1}, \cdot, \theta_{i+1}, \ldots, \theta_C)$. Let $\theta^* = \arg\min_\theta \mathrm{R}(\theta_i, \theta)$. We see that:

$$\mathrm{PR}(\theta^i) \leq \mathrm{PR}(\theta^*)$$
$$\leq \mathrm{R}(\theta_i, \theta^*) + L_z \mathcal{W}(\mathcal{D}(\theta_i), \mathcal{D}(\theta^*))$$
$$\leq \min_\theta \mathrm{R}(\theta_i, \theta) + L_z \left( \frac{1}{|\mathcal{U}|} \sum_{u \in \mathcal{U}} \mathbb{E}[\mathrm{dist}(z(u), z_{\theta^*}(u))] \right)$$
$$\leq \min_\theta \mathrm{R}(\theta_i, \theta) + L_z P_i.$$

Rewriting this, we see that:

$$\mathbb{E}_{z \sim \mathcal{D}}[\ell_i(\theta^i; z)] \leq \min_\theta \mathbb{E}_{z \sim \mathcal{D}}[\ell_i(\theta; z)] + L_z P_i.$$

If, in addition, $\ell_i$ is $\gamma$-strongly convex, then we know that:

$$L_z P_i \geq \mathbb{E}_{z \sim \mathcal{D}}[\ell_i(\theta^i; z)] - \min_\theta \mathbb{E}_{z \sim \mathcal{D}}[\ell(\theta; z)] \geq \frac{\gamma}{2} \|\theta^i - \min_\theta \mathbb{E}_{z \sim \mathcal{D}}[\ell_i(\theta; z)]\|^2.$$

Rearranging, we obtain that

$$\left\| \theta^i - \min_\theta \mathbb{E}_{z \sim \mathcal{D}}[\ell(\theta; z)] \right\|_2 \leq \sqrt{\frac{2 L_z P_i}{\gamma}}.$$

### C.4 Proof of Corollary 1

Let $\mathrm{P}$ be the performative power associated with the variables $z_\theta^{C=1}$. We first claim that the performative power of any firm in the mixture model is at most $P/C$. This follows from the fact that for a given firm the potential outcome $z_\theta(u)$ is equal to $z(u)$ with probability $1 - 1/C$ and equal to $z_\theta^{C=1}(u)$ with probability $1/C$.

Let's focus on platform $i$, fixing classifiers selected by the other platforms. Let's take $\mathrm{PR}$ and $\mathrm{R}$ to be with respect to $\mathcal{D}(\cdot) = \mathcal{D}^C(\theta^*, \cdots, \theta^*, \cdot, \theta^*, \ldots, \theta^*)$. Now, we can apply Proposition 2 to see that

$$\mathrm{PR}(\theta^*) = \mathbb{E}_{z \sim \mathcal{D}^{C=1}(\theta^*)}[\ell(\theta^*; z)] \leq \min_\theta \mathbb{E}_{z \sim \mathcal{D}^{C=1}(\theta^*)}[\ell(\theta; z)] + \frac{L_z P}{C}.$$

Thus, in the limit as $C \to \infty$, it holds that

$$\mathbb{E}_{z \sim \mathcal{D}^{C=1}(\theta^*)}[\ell(\theta^*; z)] \to \min_\theta \mathbb{E}_{z \sim \mathcal{D}^{C=1}(\theta^*)}[\ell(\theta; z)]$$

as desired.

### C.5 Proof of Lemma 1

Fix a classifier $f$ and a unit $u$. By Assumption 1, we know that $x(u)$ and $x_f(u)$ are both in $\mathcal{X}_{\Delta\gamma}(u)$. The claim follows from

$$\mathrm{dist}(x_{\mathrm{orig}}(u), x_f(u)) \leq \sup_{x' \in \mathcal{X}_{\Delta\gamma}(u)} \mathrm{dist}(x_{\mathrm{orig}}(u), x').$$

### C.6 Proof of Proposition 3

The first coordinate being immutable corresponds to $c(x, x') = \infty$ for $x[1] \neq x'[1]$. The proof is by construction of a classifier $f^* : \mathrm{R}^m \to \{0, 1\}$. For each individual $u$ we define the set

$$\tilde{\mathcal{X}}(u) := \arg\sup_{x' \in \mathcal{X}_{\Delta\gamma}(u)} \mathrm{dist}(x(u), x').$$

Now let $f^*$ be such that

$$f^*(x) = \begin{cases} 1 & x \in \tilde{\mathcal{X}}(u) \text{ with } u = x[1] \\ 0 & x \notin \tilde{\mathcal{X}}(u) \text{ with } u = x[1] \end{cases}$$

where we used that the first coordinate of the feature vector $x$ uniquely identifies the individual. The effect of $f^*$ on a population $\mathcal{U}$ corresponds to

$$\frac{1}{|\mathcal{U}|} \sum_{u \in \mathcal{U}} \text{dist}(x(u), x_{f^*}(u)) = \frac{1}{|\mathcal{U}|} \sum_{u \in \mathcal{U}} \sup_{x' \in \mathcal{X}_{\Delta\gamma}(u)} \text{dist}(x(u), x') \tag{7}$$

Thus for any $\mathcal{F}$ that contains $f^*$ the performative power is maximized.

## C.7 Proof of Corollary 2

Applying Lemma 1, it suffices to show that the diameter of the set $\mathcal{X}_{\Delta\gamma}(u)$ can be upper bounded by $2L\Delta\gamma$ for any $u \in \mathcal{U}$. We see that for any $x, x' \in \mathcal{X}_{\Delta\gamma}(u)$, using that $c$ is a metric, it holds that:

$$\begin{aligned}
\text{dist}(x, x') &\leq L \cdot c(x, x') \\
&\leq L \cdot (c(x_{\text{orig}}(u), x) + c(x_{\text{orig}}(u), x')) \\
&\leq 2L\Delta\gamma.
\end{aligned}$$

## C.8 Proof of Proposition 5

The upper bound follows from Corollary 4. For the lower bound, we take $f$ to be the threshold classifier given by $\theta_{\text{SL}} + \Delta\gamma$. We see that for $x_{\text{orig}}(u) \in [\theta_{\text{SL}}, \theta_{\text{SL}} + \Delta\gamma]$, it holds that $x_f(u) = \theta_{\text{SL}} + \Delta\gamma$ and $x(u) = x_{\text{orig}}(u)$. This means that the performative power is at least:

$$\begin{aligned}
\text{P} &= \frac{1}{|\mathcal{U}|} \sum_{u \in \mathcal{U}} \mathbb{E}[\text{dist}(x(u), x_f(u))] \\
&= \frac{1}{|\mathcal{U}|} \sum_{u \in \mathcal{U}} \mathbb{E}[|x(u) - x_f(u)|] \\
&\geq \frac{1}{|\mathcal{U}|} \sum_{u \in \mathcal{U}} I[x_{\text{orig}}(u) \in [\theta_{\text{SL}}, \theta_{\text{SL}} + \Delta\gamma]] \mathbb{E}[|\theta_{\text{SL}} + \Delta\gamma - x_{\text{orig}}(u)|] \\
&\geq \frac{1}{|\mathcal{U}|} \sum_{u \in \mathcal{U}} I[x_{\text{orig}}(u) \in [\theta_{\text{SL}}, \theta_{\text{SL}} + \frac{1}{2}\Delta\gamma]] \cdot \frac{1}{2}\Delta\gamma \\
&\geq \frac{1}{2}\Delta\gamma \underset{\mathcal{D}_{\text{orig}}}{\mathbb{P}} [x \in [\theta_{\text{SL}}, \theta_{\text{SL}} + \frac{1}{2}\Delta\gamma]],
\end{aligned}$$

as desired.

## C.9 Proof of Proposition 4

To prove this proposition, we show the following two intermediate results which are proven in the next two sections.

The proof relies on the following proposition that we will prove in the next section.

**Proposition 6.** *Consider the 1-dimensional setup specified in Section 4.3, and suppose that the economy is at a symmetric state where both firms choose classifier $\theta$. For any $\mathcal{F}$, consider one of the firms, let $\mathcal{F}$ denote their action set and let $\theta_{\min}$ be the minimum threshold classifier in $\mathcal{F}$. Then, the performative power of the firm is upper bounded by:*

$$\text{P} \leq L \min(c(\theta_{\min}, \theta), \gamma) + \gamma L p_{\text{reach}}([\theta_{\min}, \theta]).$$

*where $p_{\text{reach}}([\theta_{\min}, \theta]) := \mathbb{P}_{\mathcal{D}_{\text{orig}}} [x \in [\xi(\theta_{\min}), \xi(\theta)]]$ with $\xi(\theta')$ being the unique value such that $\xi(\theta') < \theta'$ and $c(\xi(\theta'), \theta') = 1$.*

**Proposition 7.** *Consider the 1-dimensional setup described in Section 4.3. Then, a symmetric solution $[\theta^*, \theta^*]$ is an equilibrium if and only if $\theta^*$ satisfies*

$$\mathbb{E}_{(x,y) \sim \mathcal{D}_{\text{orig}}}[y = 1 \mid x \geq \xi(\theta^*)] = \tfrac{1}{2}, \tag{8}$$

*where $\xi(\theta^*)$ is the unique value such that $c(\xi(\theta^*), \theta^*) = \gamma$ and $\xi(\theta^*) < \theta^*$. Both firms earn zero utility at this equilibrium. Moreover, the set $\mathcal{F}^+(\theta^*)$ of actions that a firm can take at equilibrium that achieve nonnegative utility is exactly equal to $[\theta^*, \infty)$, assuming the other firm chooses the classifier $\theta^*$.*

We now prove Proposition 4 from these intermediate results. We apply Proposition 6 to see that the performative power is upper bounded by

$$B := L\min(c(\theta_{\min}, \theta^*), \gamma) + L\gamma p_{\text{reach}}([\theta_{\min}, \theta^*])$$

where $(\theta^*, \theta^*)$ is a symmetric state. Using Proposition 2, we see that $\mathcal{F}(\theta^*) = [\theta^*, \infty)$. This means that $\theta_{\min} = \theta^*$, and so $\xi(\theta_{\min}) = \xi(\theta^*)$. Thus, $B = 0$ which demonstrates that the performative power is upper bounded by $0$, and is thus equal to $0$.

### C.10  Proof of Proposition 6

Consider a classifier $f \in \mathcal{F}(\theta)$ with threshold $\theta'$, and suppose that a firm changes their classifier to $f$. It suffices to show that:

$$\frac{1}{|\mathcal{U}|} \sum_{u \in \mathcal{U}} \mathbb{E}[\text{dist}(x(u), x_f(u))] \leq L\min(c(\theta_{\min}, \theta), \gamma) + L\gamma p_{\text{reach}}([\theta_{\min}, \theta]).$$

For technical convenience, we reformulate this in terms of the cost function $c$. Based on the definition of $L$, it suffices to show that:

$$\frac{1}{|\mathcal{U}|} \sum_{u \in \mathcal{U}} \mathbb{E}[c(x(u), x_f(u))] \leq \min(c(\theta_{\min}, \theta), \gamma) + \gamma p_{\text{reach}}([\theta_{\min}, \theta]).$$

**Case 1: $\theta' > \theta$.** Participants either are indifferent between $\theta$ and $\theta'$ or prefer $\theta$ to $\theta'$. Due to the tie breaking rule, the firm will thus lose all of its participants. Thus, all participants will switch to the other firm and adapt their features to that firm which has threshold $\theta$. This is the same behavior as these participants had in the current state, so $x_f(u) = x(u)$ for all participants $u$. This means that

$$\frac{1}{|\mathcal{U}|} \sum_{u \in \mathcal{U}} \mathbb{E}[c(x(u), x_f(u))] = 0$$

as desired.

**Case 2: $\theta' < \theta$.** Participants either are indifferent between $\theta$ and $\theta'$ or prefer $\theta'$ to $\theta$. Due to the tie breaking rule, the firm will thus gain all of the participants. We break into several cases:

$$\begin{cases} x_f(u) = x(u) = x_{\text{orig}}(u) & \text{if } x_{\text{orig}}(u) < \xi(\theta') \\ x_f(u) = \theta', x(u) = x_{\text{orig}}(u) & \text{if } x_{\text{orig}}(u) \in [\xi(\theta'), \min(\theta', \xi(\theta)))] \\ x_f(u) = x(u) = x_{\text{orig}}(u) & \text{if } x_{\text{orig}}(u) \in (\theta', \xi(\theta)) \\ x_f(u) = \theta', x(u) = \theta & \text{if } x_{\text{orig}}(u) \in (\xi(\theta), \theta') \\ x_f(u) = x_{\text{orig}}(u), x(u) = \theta & \text{if } x_{\text{orig}}(u) \in [\max(\theta', \xi(\theta)), \theta] \\ x_f(u) = x(u) = x_{\text{orig}}(u) & \text{if } x_{\text{orig}}(u) \geq \theta. \end{cases}$$

The only cases that contribute to $\frac{1}{|\mathcal{U}|} \sum_{u \in \mathcal{U}} \mathbb{E}[c(x(u), x_f(u))]$ are the second, fourth, and fifth cases. Thus, we can upper bound $\frac{1}{|\mathcal{U}|} \sum_{u \in \mathcal{U}} \mathbb{E}[c(x(u), x_f(u))]$ by:

$$\underbrace{\frac{1}{|\mathcal{U}|} \sum_{u \in \mathcal{U} | x_{\text{orig}}(u) \in [\xi(\theta'), \min(\theta', \xi(\theta))]} \mathbb{E}[c(x(u), x_f(u))]}_{(A)} + \underbrace{\frac{1}{|\mathcal{U}|} \sum_{u \in \mathcal{U} | x_{\text{orig}}(u) \in (\xi(\theta), \theta')} \mathbb{E}[c(x(u), x_f(u))]}_{(B)}$$

$$+ \underbrace{\frac{1}{|\mathcal{U}|} \sum_{u \in \mathcal{U} | x_{\text{orig}}(u) \in [\max(\theta', \xi(\theta)), \theta]} \mathbb{E}[c(x(u), x_f(u))]}_{(C)}$$

For (A), we see that

$$(A) = \frac{1}{|\mathcal{U}|} \sum_{u \in \mathcal{U} | x_{\text{orig}}(u) \in [\xi(\theta'), \min(\theta', \xi(\theta)))} \mathbb{E}[c(x_{\text{orig}}(u), \theta')]$$

$$\leq \frac{1}{|\mathcal{U}|} \sum_{u \in \mathcal{U} | x_{\text{orig}}(u) \in [\xi(\theta'), \min(\theta', \xi(\theta)))} \mathbb{E}[c(\xi(\theta'), \theta')]$$

$$= \gamma \cdot \mathbb{P}_{\mathcal{D}_{\text{orig}}}[x \in [\xi(\theta'), \min(\theta', \xi(\theta))))]$$

$$\leq \gamma \cdot \mathbb{P}_{\mathcal{D}_{\text{orig}}}[x \in [\xi(\theta'), \xi(\theta))].$$

For (B), we see that:

$$(B) = \frac{1}{|\mathcal{U}|} \sum_{u \in \mathcal{U} | x_{\mathrm{orig}}(u) \in (\xi(\theta), \theta')} \mathbb{E}[c(\theta, \theta')]$$
$$= c(\theta, \theta') \cdot \mathbb{P}_{\mathcal{D}_{\mathrm{orig}}}[x \in (\xi(\theta), \theta')]$$
$$= \min(c(\theta, \theta'), \gamma) \cdot \mathbb{P}_{\mathcal{D}_{\mathrm{orig}}}[x \in (\xi(\theta), \theta')].$$

For (C), we see that:

$$(C) = \frac{1}{|\mathcal{U}|} \sum_{u \in \mathcal{U} | x_{\mathrm{orig}}(u) \in [\max(\theta', \xi(\theta)), \theta]} \mathbb{E}[c(x_{\mathrm{orig}}(u), \theta)]$$
$$\leq \frac{1}{|\mathcal{U}|} \sum_{u \in \mathcal{U} | x_{\mathrm{orig}}(u) \in [\max(\theta', \xi(\theta)), \theta]} \mathbb{E}[\min\left(c(\theta', \theta), c(\xi(\theta), \theta)\right)]$$
$$= \frac{1}{|\mathcal{U}|} \sum_{u \in \mathcal{U} | x_{\mathrm{orig}}(u) \in [\max(\theta', \xi(\theta)), \theta]} \mathbb{E}[\min\left(c(\theta', \theta), \gamma\right)]$$
$$= \min(c(\theta', \theta), \gamma) \cdot \mathbb{P}_{\mathcal{D}_{\mathrm{orig}}}[x \in [\max(\theta', \xi(\theta)), \theta]]$$

Putting this all together, we obtain that:

$$\frac{1}{|\mathcal{U}|} \sum_{u \in \mathcal{U}} \mathbb{E}[c(x(u), x_f(u))] \leq \gamma p_{\mathrm{reach}}([\theta', \theta]) + \min(c(\theta', \theta), \gamma)$$

for $p_{\mathrm{reach}}([\theta', \theta]) := \mathbb{P}_{\mathcal{D}_{\mathrm{orig}}}[x \in [\xi(\theta'), \xi(\theta)]]$ as desired. Since $\gamma p_{\mathrm{reach}}([\theta', \theta]) + \min(c(\theta', \theta), \gamma)$ is decreasing in $\theta'$, this expression is maximized when $\theta' = \theta_{\mathrm{min}}$. Thus we obtain an upper bound of

$$\gamma \cdot p_{\mathrm{reach}}([\theta_{\mathrm{min}}, \theta]) + \min(c(\theta_{\mathrm{min}}, \theta), \gamma).$$

### C.11 Proof of Proposition 7

The proof proceeds in two steps. First, we establish that $[\theta^*, \theta^*]$ is an equilibrium; next, we show that $[\theta, \theta]$ is not in equilibrium for $\theta \neq \theta^*$.

**Establishing that $[\theta^*, \theta^*]$ is an equilibrium and $\mathcal{F}^+(\theta^*) = [\theta^*, \infty)$.** First, we claim that $[\theta^*, \theta^*]$ is an equilibrium. At $[\theta^*, \theta^*]$, each participant chooses the first firm with $1/2$ probability. The expected utility earned by a firm is:

$$\frac{1}{2} \int_{\xi(\theta)}^{\infty} p_{\mathrm{orig}}(x)(p(x) - (1 - p(x))) \mathrm{d}x = \int_{\xi(\theta)}^{\infty} p_{\mathrm{orig}}(x)(p(x) - 0.5) \mathrm{d}x$$
$$= \int_{\xi(\theta)}^{\infty} p_{\mathrm{orig}}(x) p(x) \mathrm{d}x - 0.5 \int_{\xi(\theta)}^{\infty} p_{\mathrm{orig}}(x) \mathrm{d}x$$
$$= \int_{\xi(\theta)}^{\infty} p_{\mathrm{orig}}(x) \mathrm{d}x \left( \frac{\int_{\xi(\theta)}^{\infty} p_{\mathrm{orig}}(x) p(x) \mathrm{d}x}{\int_{\xi(\theta)}^{\infty} p_{\mathrm{orig}}(x) \mathrm{d}x} - \frac{1}{2} \right)$$
$$= \left( \int_{\xi(\theta)}^{\infty} p_{\mathrm{orig}}(x) \mathrm{d}x \right) \left( \mathop{\mathbb{E}}_{(x,y) \sim \mathcal{D}_{\mathrm{orig}}} [y = 1 \mid x \geq \xi(\theta)] - \frac{1}{2} \right)$$
$$= 0.$$

If the firm chooses $\theta > \theta^*$, then since the cost function is strictly monotonic in its second argument, participants either are indifferent between $\theta$ and $\theta^*$ or prefer $\theta$ to $\theta^*$. Due to the tie breaking rule, the firm will thus lose all of its participants and incur 0 utility. Thus the firm has no incentive to switch to $\theta$.

If the firm chooses $\theta < \theta^*$, then it will gain all of the participants. The firm's utility will be:

$$\int_{\xi(\theta)}^{\infty} p_{\text{orig}}(x)(p(x) - (1 - p(x)))\mathrm{d}x$$

$$= \int_{\xi(\theta)}^{\xi(\theta^*)} p_{\text{orig}}(x)(p(x) - (1 - p(x)))\mathrm{d}x + \int_{\xi(\theta^*)}^{\infty} p_{\text{orig}}(x)(p(x) - (1 - p(x)))\mathrm{d}x$$

$$= 2\int_{\xi(\theta)}^{\xi(\theta^*)} p_{\text{orig}}(x)(p(x) - 0.5)\mathrm{d}x.$$

It is not difficult to see that at $\theta^*$, it must hold that $p(\xi(\theta^*)) \leq 0.5$. Since the posterior is strictly increasing, this means that $p(\xi(\theta)) < p(\xi(\theta^*)) = 0.5$, so the above expression is negative. This means that the firm will not switch to $\xi(\theta)$.

Moreover, this establishes that $\mathcal{F}(\theta*) = [\theta^*, \infty)$.

$[\theta, \theta]$ **is not in equilibrium if** $\xi(\theta^*)$ **does not satisfy** (8). If $\theta < \theta^*$, then the firm earns utility

$$\frac{1}{2}\left(\int_{\xi\theta}^{\infty} p_{\text{orig}}(x)(p(x) - (1 - p(x)))\right)\mathrm{d}x,$$

which we already showed above was negative. Thus, the firm has incentive to change their threshold to above $\theta$ so that it loses the full participant base and gets 0 utility.

If $\theta > \theta^*$, then the firm earns utility

$$U = \frac{1}{2}\left(\int_{\xi(\theta)}^{\infty} p_{\text{orig}}(x)(p(x) - (1 - p(x)))\right)\mathrm{d}x,$$

which is strictly positive. Fix $\epsilon > 0$, and suppose that the firm changes to a threshold $\theta'$ such that $c(\theta', \theta) = \epsilon$. Then it would gain all of the participants and earn utility:

$$\int_{\xi(\theta')}^{\infty} p_{\text{orig}}(x)(p(x) - (1 - p(x)))\mathrm{d}x = \int_{\xi(\theta')}^{\xi\theta} p_{\text{orig}}(x)(p(x) - (1 - p(x)))\mathrm{d}x$$

$$+ \int_{\xi(\theta)}^{\infty} p_{\text{orig}}(x)(p(x) - (1 - p(x)))\mathrm{d}x$$

$$= \int_{\xi(\theta')}^{\xi(\theta)} p_{\text{orig}}(x)(p(x) - (1 - p(x)))\mathrm{d}x + 2U.$$

We claim that this expression approaches $2U$ as $\epsilon \to 0$. To see this, note that $c(\xi(\theta'), \theta) \to \gamma$ and so $\xi(\theta') \to \xi(\theta)$ as $\epsilon \to 0$. This implies that $\int_{\xi(\theta')}^{\xi(\theta)} p_{\text{orig}}(x)(p(x) - (1 - p(x)))\mathrm{d}x \to 0$ as desired. Thus, the expression approaches $2U > U$ as desired. This means that there exists $\epsilon$ such that the firm changing to $\theta'$ results in a strict improvement in utility.

## C.12 Proof of Theorem 1

Recall the definition of the action set $\mathcal{S}$. We prove Theorem 1 by constructing a $s_{\text{swap}} \in \mathcal{S}$ and relating the effect of a change in the score function from $s_{\text{curr}}$ to $s_{\text{swap}}$ to the causal effect of position.

For $u \in \mathcal{U}$ let $i_{\text{top}}(u)$ and $i_2(u)$ denote the index of the content item shown to user $u$ under $s_{\text{curr}}$ in the first and second display slot, respectively. Now, let the score function $s_{\text{swap}}$ be such that the content displayed in the first two display slots is swapped relative to $s_{\text{curr}}$, simultaneously for all users $u \in \mathcal{U}$:

$$\tilde{s}(u)[i] = \begin{cases} s_{\text{curr}}(u)[i_2(u)] & i = i_{\text{top}}(u) \\ s_{\text{curr}}(u)[i_{\text{top}}(u)] & i = i_2(u) \\ s_{\text{curr}}(u)[i] & \text{otherwise.} \end{cases} \tag{9}$$

It holds that $s_{\text{swap}} \in \mathcal{S}$, since $|s_{\text{curr}}(u)[i_{\text{top}}(u)] - s_{\text{curr}}(u)[i_2(u)]| \leq \delta$ for all $u \in \mathcal{U}$. We lower bound performative power as

$$\mathrm{P} = \sup_{s \in \mathcal{S}} \frac{1}{|\mathcal{U}|} \sum_{u \in \mathcal{U}} \mathbb{E}\left[\|z(u) - z_s(u)\|_1\right] \geq \frac{1}{|\mathcal{U}|} \sum_{u \in \mathcal{U}} \mathbb{E}\left[\|z(u) - z_{s_{\text{swap}}}(u)\|_1\right] \tag{10}$$

To bound the difference between the counterfactual variable $z(u)$ and $z_{s_{\mathrm{swap}}}(u)$, we decompose $s_{\mathrm{swap}}$ into a series of unilateral *swapped* score functions, one for each viewer. The score function $s_{\mathrm{swap}}^u$ associated with viewer $u$ swaps the scores of content that currently appears in the first two display slots for viewer $u$ and keeps the scores of the other viewers unchanged.

Assumption 2 implies that $z_{s_{\mathrm{swap}}}(u) = z_{s_{\mathrm{swap}}^u}(u)$, since there are no peer effects; $z_{s_{\mathrm{swap}}}(u)$ is independent of $s_{\mathrm{swap}}(u')$ for $u' \neq u$. Thus, we can aggregate the unilateral effects across all viewers $u \in \mathcal{U}$ to obtain the effect of $s_{\mathrm{swap}}$ as:

$$\mathrm{P} \geq \frac{1}{|\mathcal{U}|} \sum_{u \in \mathcal{U}} \mathbb{E}\left[\|z(u) - z_{s_{\mathrm{swap}}}(u)\|_1\right] = \frac{1}{|\mathcal{U}|} \sum_{u \in \mathcal{U}} \mathbb{E}\left[\|z(u) - z_{s_{\mathrm{swap}}^u}(u)\|_1\right] \tag{11}$$

Reasoning about unilateral effects allows us to relate the summands in (11) to the causal effect of position. In particular, focus on coordinate $i_{\mathrm{top}}(u)$ in the norm, and let $Y_0(u) = z(u)[i_{\mathrm{top}}(u)]$ and $Y_1(u) = z_{s_{\mathrm{swap}}^u}(u)[i_{\mathrm{top}}(u)]$. Then, we have

$$\mathrm{P} \geq \frac{1}{|\mathcal{U}|} \sum_{u \in \mathcal{U}} \mathbb{E}\left|z(u)[i_{\mathrm{top}}(u)] - z_{s_{\mathrm{swap}}^u}(u)[i_{\mathrm{top}}(u)]\right| = \frac{1}{|\mathcal{U}|} \sum_{u \in \mathcal{U}} \mathbb{E}\left|Y_0(u) - Y_1(u)\right| = \beta.$$

where the causal effect of position $\beta$ is defined as in Definition 2.