# OpenReview forum: "Performative Power"
_NeurIPS.cc/2022/Conference — NeurIPS 2022 Accept_

### Official Review · Reviewer_rFMx · 2022-07-08

**Rating:** 6
**Confidence:** 3
**Soundness:** 4 excellent
**Presentation:** 4 excellent
**Contribution:** 3 good

**Summary:**

The paper proposes a new measure of the power of a firm in a market, namely the performative power.  The idea is the firm's actions can affect the behavior of the participants of the market, causing them to deviate from the "natural" state without the presence of the firm.  And the performative power of the firm is the maximum average deviation it can cause by performing some action.  The authors show how this notion connects to a number of issues, including (1) whether a firm should focus on learning customers' preferences stastistically or changing their preferences to align better with the firm's goal, and (2) how much benefit the firm can achieve by optimizing the objective post the response, vs optimizing statistically without considering the effect of the response.  They also show how to bound the performative power in some concrete settings.

**Questions:**

(Also putting some detailed comments here.)

Line 40, "a second way": any evidence that this actually happens?  And to what extent?  I know this might be hard to measure -- just curious.

Paragraphs between line 114 and line 130: I find part of the argument a bit shaky.  First the authors say that the actions that we should consider are those which "increase profit for the firm".  It seems this should really be interpreted as actions that "align with the objective of the firm".  But later, the authors argue that it's "not necessary to know the firm's objective function ..."  I wonder how the two claims should be reconciled.

Line 216: "let \phi by any ..."

Sec 4.2, \gamma vs \Delta \gamma: while this is conceptually interesting, I wonder how much technical difference there is when \gamma is replaced by \Delta \gamma.  If I understand it correctly, one can simply replace every appearance of \Delta \gamma in sec 4.2 with \gamma, and then we are back in the world where conceptually there isn't an outside option?  In that case I wouldn't overemphasize the effect of outside options.  Conceptually, the outside option formulation also implicitly assumes that a participant can always get utility \beta if they choose so, no matter what their features are.  This in a sense creates asymmetry between the "inside" option and the outside option (one has to "qualify" to get the inside utility, but can get the outside utility no matter what).  So perhaps the outside option is more like unemployment benefits...

Presentation in general: the paper is generally quite well written and polished, but a small thing that I think can be improved is the (somewhat unnecessary) use of jargon.  Some examples: I'd replace "performativity" with "best response" or something like that; in line 363, it's not clear what "standard microfoundations" are and why we need them if the reader is not familiar with "alternative microfoundations", and it seems totally fine to just say "Assumption 1".  Overall I think the presentation of the paper is heavily influenced by a particular line of work (which is of course fine), but maybe the paper would be more enjoyable for the general audience if presented in a more self-contained way.

**Limitations:**

Overall I'm satisfied (see detailed comments above for some minor suggestions).

**Strengths And Weaknesses:**

Strengths

The contributions of the paper are predominantly conceptual, so my evaluation is inevitably subjective.  I like the idea of measuring the power of a firm in a clean and well-defined way.  In particular, I think it's very nice that the definition doesn't rely too much on the specific form of the market (we still need some prior knowledge in order to properly choose the set of actions, the dist function, etc.).  The authors are able to confirm some natural phenomenon within the new framework and draw connections to other quantities of interest, which makes the new measure more credible.  They also show how to bound the measure (albeit in somewhat simple settings), which makes it more useful.


Weaknesses

Although the new measure definitely makes sense in some ways, it's not immediately clear to me it's the best way to measure the potential harm a firm can do to the market.  One potential criticism is that the measure is too pessimistic, and a rational and profit-driven firm wouldn't actually cause nearly as much harm.  Also, the authors present a number of applications of the new measure, but I'd be more excited if there's a "killer" application where the new measure helps solve a real problem that was previously beyond reach (it doesn't even have to be a technical one).  Also see detailed comments for some minor points.

---

> ### Author Response · Authors · 2022-08-02
> **Response to Reviewer rFMx**
>
> **On the firm steering users to be easier to monetize**: We first highlight that this form of steering can arise even without explicit consideration by the firm. For example, the firm employs A/B testing to evaluate profit of candidate models, this process would guide the firm towards models that have high ex-post profit, e.g. through steering the population towards patterns that are easier to monetize. Anecdotally, steering could be manifested by firms showing users extreme content that shapes their preferences, through behavioral modifications that reamplify the validity of predictions, or by driving consumption towards products of their own label. We hope that our definition and framework inspire future work to perform more thorough empirical assessments of the ways in which steering arises in digital marketplaces.
>
> **On applications of performative power**: In the paper, we present two applications of performative power: (a) an empirical approach for measuring the power of the platform to steer user consumption patterns in recommender systems (Display Discontinuity Design)  and (b) a theoretical analysis of strategic classification (Section 4). We hope that our definition helps formalize the forms of power that can be exhibited by digital platforms, and future work pinpoints further applications of interest.
>
> **On the introduction of variants of strategic classification**: Our primary motivation for introducing variants of strategic classification was to build on a standard setup, familiar to many researchers in the NeurIPS community, to illustrate the behavior of performative power in different market contexts. Moreover we believe it is valuable to pinpoint some implicit economic assumptions underlying the current framework. We do agree that our extensions of strategic classification are stylized and simple, but they serve the purpose to illustrate the qualitative behavior of performative power with different economic factors. Certainly, developing richer extensions of strategic classification would be an interesting avenue for future work.
>
> **On the presentation of the paper**: We will incorporate further clarifications for terms such as “performative” and “microfoundations”. We do wish to emphasize, though, that performative effects can arise not only from strategic manipulations to features (e.g. best-response), but also from changes to the labels. For example, see the example of self-fulfilling prophecies in Section 2.1.

---

> > ### Comment · Reviewer_rFMx · 2022-08-08
> > **Thanks for the reponse**
> >
> > The response does answer my questions (which are meant to be constructive rather than critical).  Given the detailed response, I'm sure the authors can incorporate all 3 reviews (also using their own judgment) and further improve the paper.  At the moment, I don't have further questions.

---

### Official Review · Reviewer_BiVM · 2022-07-09

**Rating:** 8
**Confidence:** 3
**Soundness:** 3 good
**Presentation:** 3 good
**Contribution:** 3 good

**Summary:**

The authors study the degree to which the deployment of predictors can influence, or cause changes in, a population interacting with said predictor.  This paradigm is framed through the lens of recommender systems and strategic classification in which a firm deploys a model $f$ which makes decisions on a population $U$, where each $u\in U$ has original data $z(u)$ and modified data $z_f(u)$ in the presence of model $f$. A firm's ability to shift a population is formalized in the proposed definition of Performative Power which measures the largest average deviation to agents' data that the firms choice of $f$ can have.  Several theoretical results are shown such as demonstrating how performative power changes with and without competition, as well as linking  performative power to the well established notion of performative prediction.

**Questions:**

- **Framing of performative power**: Throughout the paper performative power is framed as a potentially negative phenonium, meaning we may want to be wary of firms with large performative power. While this certainly seems true, there is also be a positive side to performative power which appears to be absent from the paper. For example, in the context of lending, agents can manipulate their features to appear more creditworthy to the firm, and through the lens of Strategic Classification, it is assumed that these are non-productive manipulations such as lying on applications (i.e. an agents true creditworthiness is not effected). However, some "manipulations" may be productive in the sense that not only do they cause the agent to appear more credit worthy, but they also cause the agent to actually be more creditworthy. For example, decreasing one's debt and closing excessively opened credit cards will have both effects. In a setting like this, where the alterations to an agents features have a causal relationship on their true label in a "positive" way, the performative power of a firm is desirable in the sense that the firm could help induce stronger financial literacy and true creditworthiness throughout the population.


- **Definition 1**: Performative power is defined via supremum over the firm's action space $\mathcal{F}$. While I understand the motivation for the supremum, I am curious how the results or interpretations change when performative power is defined in terms of an average over $\mathcal{F}$. Are we missing anything of value by using a supremum instead of an average?

- **Proposition 2**: How reliant is Proposition 2 on the fact that the classifiers from each firm are thresholds? It seems that when we move away from thresholds to more complex models, the idea of  threshold predictors can still somewhat apply in the sense that  $x$ can be thought of as the outcome of the classifier's score function $h$, and we are now thresholding on that score. However, in this setting the cost function should now depend on the underlying features, not on the model's scores, and each firm now has the option to choose both the score function and the associated threshold . It is unclear to me how to extend this result past the 1-D case. Can we say anything for more complex models?


- **Minor comments/issues**:

    - Line 102: It may help readability to specify the domain of $z(u)$. For example, does $z(u)$ contain only predictive features? Is there a label associated with $z(u)$. This is cleared up later in the paper, but referring to $z(u)$ simply as a data point may be too vague for some readers.

    - Line 102: Using $d$ instead of $\text{dist}$ may be more in line the notation of prior literature.




**Limitations:**

The authors clearly outline their limitations and are overtly transparent about the applicableness of their work.

**Strengths And Weaknesses:**

## Strengths

- This paper is well placed in the literature and lays a solid foundation for an interesting problem. The idea of performative power is an insightful extension of performative prediction and opens many directions for future work.

- The paper is extremely well written.

- The authors provide several strong theoretical results which serve to outline the types of questions/observations one may expect to see in the setting of performative power.

- Intuition and interpretation for each result is provided.

- Example 2.1 provides both a concrete way to understand the problem, as well as a highly topical and realistic setting which the authors work is directly applicable.

- Section 4 demonstrates that performative power is relevant  in several well studied fields/settings.

- Performative power is examined in both isolation (single firm) and competition (multiple firms).

## Weaknesses

- Proofs being deferred to the supplement can cause parts of the paper to feel incomplete. I would have liked to see at least a proof sketch or some intuitive reason as to why the results hold. Some results, such as Theorem 1 have an intuitive explanation preceding their formal statement, but I would still prefer to see proof sketches.

- Similar to the comment of proofs being deferred to the supplement, I think the section on display discontinuity design could benefit from more details. My first thought when looking at Definition 1 (as pointed out on line 131) is that performative power is likely to be intractable, or impossible, to measure in any type of realistic setting. The idea of display discontinuity design lends credence to performative power not only as a worthwhile metric and analytical tool, but as something that may be feasible to examine in practice. With that said I understand that at a certain point things must be deferred to the supplement, and this paper covers a lot of material so I'm sure the choice of what to defer is not straightforward.

- There is a slight disconnect between the definition of Performative Power and some of the motivation/examples given by the authors. Specifically, in the context of a firm steering a population in a desired direction. The notion of performative power measures the worst case average shift that a firm can induce in a population. This shift is measured in terms of magnitude only. I understand that results such as Theorem 1 are trying to tie performative power to a firms ability to steer the data in a desired direction, but there should perhaps be some clarification that performative power does not necessarily, or directly, measure a firm's ability to steer a population in a desired direction. In the language of Adversarial Machine Learning, the current definition of performative power takes an "untargeted" perspective. Perhaps examining performative power from a "targeted" perspective is also worth investigating.

---

> ### Author Response · Authors · 2022-08-02
> **Response to Reviewer BiVM**
>
> **Presentation**: We thank the reviewer for the suggestion on the presentation; we will make use of the additional content page to incorporate these in a final version and expand on the display discontinuity design in the main body.
>
> **On productive manipulations**: The setting of strategic classification where feature changes improve outcomes is an interesting special case where steering benefits users. However this argument crucially relies on alignment between the firm’s objective and the user’s objective. Without perfect alignment, it is difficult to guarantee that steering benefits users. For this reason, we take a user utility-agnostic approach and capture all instances where the firm can steer the population. See the general response (“Distinction between power and harm”) for a discussion. We will work on the presentation to make sure ‘power’ and ‘harm’ are more explicitly kept apart in the paper.
>
> **On the supremum over actions**:  For some further intuition, the set $\mathcal{F}$  captures the set of reasonable actions that the firm might take in practice. For example, in the display discontinuity design, this includes perturbations of the current action. We take a supremum since we wish to capture the potential for a firm to steer users: see the general response (“Potential for changes to users versus manifested changes to users”) for a discussion.
>
> **On the 1-dimensional setup in Proposition 2**: The claim of Proposition 2 is indeed limited to one-dimension, as it aims to establish a correspondence between the threshold chosen by the classifier and the price considered in Bertrand competition models in economics. In high dimensions, we envision that competition would still drive down the amount of strategic manipulation exhibited by positively labeled users. However, the preference relationship of users over actions would inevitably be more complex in high dimensions—formalizing this would be an interesting direction for future work.

---

> > ### Comment · Reviewer_BiVM · 2022-08-08
> > **Response to Authors**
> >
> > Thank you for the clarification. The intuition for the supremum makes sense, especially when $\mathcal{F}$ is intended to capture only "reasonable" actions the firm may take. With respect to this set though, I wish to echo what Reviewer hRaw has said, in that I too would find it helpful to see more discussion on the actual selection $\mathcal{F}$.
> >
> > The distinction on "harm" vs "power" is also helpful. Based on the issue of alignment between the firm and the user it seems reasonable to take a user utility-agnostic approach. Hopefully in the future we can see some results about aligned objectives.
> >
> > After reading the reviews, and the authors' responses to these reviews, I stand by my initial recommendation. This is an interesting and well written paper which covers a very important problem.

---

### Official Review · Reviewer_hRaw · 2022-07-11

**Rating:** 7
**Confidence:** 4
**Soundness:** 3 good
**Presentation:** 4 excellent
**Contribution:** 3 good

**Summary:**

The paper proposes and formalizes a novel framing to understand a decision maker’s ability to affect the behavior of the population it is making decisions on. The paper relates this notion of “performative power” to classical notions in the economics literature: they draw parallels with the notions of “market power”, of being a price-taker vs a price-maker or a monopoly, and with Bertrand competitions in competitive settings.

**Questions:**

- Can the authors provide more examples of standard economic/learning settings with strategic agents and relate the performative power there to the actual ability to steer the population in a beneficial way? The performative power is defined to be best-case and it would be useful to understand practically when this best case is realistic.

**Limitations:**

The limitations of the paper seem discussed. In particular, the second paragraph of Section 5 takes potential harms into account, and echoes some of the weaknesses discussed in this review (performative power measures *potential* harm or benefits from affecting population behavior, not necessarily *realized* harms/benefits.)

**Strengths And Weaknesses:**

- The main strength of the paper comes in the framing. Using a statistical measure like performative power abstracts away from market specifics. It is useful in the sense that modelling the specifics of a given problem often gives rise to complex problems with intractable equilibria, versus performative power can be measured from simply looking at the outcomes of said markets.
- The notion of performative power is a measure of/proxy for a firm’s ability to affect its users. By extension, it provides some useful understanding of when it is vs is not possible to benefit from affecting the target population’s behavior.
- The parallels with economics are nice and useful to understand the value of the framework. I think the monopoly and the Bertrand competition are especially nice settings where the notion of performative power is “tight”: competition implies 0 performative power, meaning the firms have no influence on the populations and might as well solve the ex-ante problem, now equivalent to the ex-post one; for monopolies, the performative power is closely bounded by the surplus utility.


Weaknesses:
- Overall, the main theorem remains fairly weak. Theorem 1 seems almost immediate given the Lipschitzness assumption, and it only gives an upper bound on the difference between the performances of \theta_PO and \theta_SL. In turn, it says that with low performative power, the performance of \theta_PO is unlikely to me much better than \theta_SL, given that we have little power to steer the population hence the ex-ante and ex-post optimization do not differ by much anyway. It does not however say that high performative power guarantees significant differences in performance between \theta_PO and \theta_SL.
- Related to the point above: bcause the notion of performative power looks at the supremum over possible actions/decision rules f deployed by the firm, higher performative power seems to only guarantee that the population can be steered in some, not all cases. In turn, it is not clear that performative power means that you can steer the population *in a way that is beneficial to the firm*. It would be nice to have more insights on situations in which perfomative power is “tight” in that sense.

Overall, I think the approach and setting of the paper are valuable and practically useful. As such, I think this paper would be a good addition to the program and spark discussion. However, I think the results could be pushed further and it may be worth illustrating the power (pun intended) of the framework a bit more.

---

> ### Author Response · Authors · 2022-08-02
> **Response to Reviewer hRaw**
>
> **Connection to standard economic and learning examples**: Apart from the strategic classification setting in Section 4, another setting that elucidates the behavior of performative power is a traditional market for homogeneous goods. Suppose that the firm actions are taken to be prices, and suppose that the data $z(u)$ for a user corresponds to an indicator for whether the user purchases the good. A simple calculation shows that in a monopoly setting, performative power is nonzero, whereas in a Bertrand competition setting, performative power is 0. This captures the classical economic intuition that 2 firms is sufficient to drive the market power to 0 in the Bertrand competition model, demonstrating the close connection between performative power and market power.
>
> **When is performative power best-case?** See the general response (“Potential for changes to users versus manifested changes to users”) for a discussion of how performative power is largely objective-agnostic. Nonetheless, one situation where the firm is incentivized to steer the population and steering increases user burden is illustrated in the strategic classification example in Section 4. We show that in the monopoly setting, a firm whose objective corresponds to accuracy would wish to set the threshold to the Stackelberg equilibrium which inevitably makes the individuals around the supervised learning thresholds exert maximal effort to maintain a positive prediction.

---

> > ### Comment · Reviewer_hRaw · 2022-08-07
> > **Thanks for the response!**
> >
> > Thanks for the useful comments! I think it'd be nice to add a bit more discussion to the order of the second point in the general response on restricting $\mathcal{F}$ and on guidance on how to do so. Overall, this gives an interesting direction to address my main question/concern, so I'm increasing my score to a 7/would like to see the paper be part of the program regardless.

---

### Author Response · Authors · 2022-08-02
**General response to all reviewers**


Thank you to the reviewers for their thoughtful feedback and appreciation of our work! We provide a general response below and then respond to each reviewer individually.

**Distinction between power and harm**: Several reviewers raised interesting comments about the implications of our work for quantifying the harm that can be inflicted by a platform on users. We emphasize that our goal is to formalize power (“ability to direct or influence the behavior of others”) rather than harm. While harm would capture the negative consequences on users, power quantifies the capacity to exert influence on users. In particular, this means that performative power is agnostic to the specifics of user utility functions, whereas harm inevitably depends on the assumption about these utility functions. From a practical perspective, we envision that performative power could serve as a tool to flag market contexts that merit further investigation by regulators, rather than as proof of negative consequences. This is akin to the difference between a firm having high market power (e.g., high market share) in classical economies, and the firm actually abusing it to raise prices on the expense of user welfare.

**Potential for changes to users versus manifested changes to users**: Several reviewers shared interesting thoughts about whether or not the platform is incentivized to take actions that steer users. Our motivation for measuring the potential for a firm to steer users over an abstract set $\mathcal{F}$ is that we do not want to embed the firm’s objective (which could be complex, partially unknown to the regulator and change over time) into the definition. Instead the choice of $\mathcal{F}$ is left up to the regulator based on knowledge of the particular use-case.

A feasible approach to close the gap between potential and manifested changes, is to restrict the action set F. For example, this could be taken to be actions that a firm has taken in the past, and performative power could be measured by performing causal inference over historical data. As another example, in the display discontinuity design we only consider small perturbations of the current action f to obtain a measure that is likely to potentially manifest; indeed, this set of action suffices to show that a platform can steer consumption patterns through how it arranges content into slots. Finally, in our theoretical analysis in Section 4.2, we choose the set to only contain profit increasing actions: this additional restriction on $\mathcal{F}$ gives us a tighter measure for this special case.

---

### Meta-Review · Area_Chair_qgdz · 2022-08-23

**Recommendation:** Accept
**Confidence:** Certain

**Metareview:**

The paper proposes the notion of "performative power" to measure a firm's ability to affect its users. This notion is insightful in performative prediction, and the authors have demonstrated it in multiple concrete settings. Overall, all reviewers are very positive about the paper. We recommend including the paper in the program and  believe it could open up potential future research directions.

**Award:**

No

---

### Decision · Program_Chairs · 2022-09-14

Accept